# REGRESSION PRIOR NETWORKS

## ABSTRACT

Prior Networks are a class of models which yield interpretable measures of uncertainty and have been shown to outperform state-of-the-art ensemble approaches on a range of tasks. They can also be used to distill an ensemble of models via *Ensemble Distribution Distillation* (EnD$^2$), such that its accuracy, calibration, and uncertainty estimates are retained within a single model. However, Prior Networks have so far been developed only for classification tasks. This work extends Prior Networks and EnD$^2$ to regression tasks by considering the Normal-Wishart distribution. The properties of Regression Prior Networks are demonstrated on synthetic data, selected UCI datasets, and two monocular depth estimation tasks. They yield performance competitive with ensemble approaches.

## 1 INTRODUCTION

Neural Networks have become the standard approach to addressing a wide range of machine learning tasks (Girshick, 2015; Simonyan & Zisserman, 2015; Villegas et al., 2017; Mikolov et al., 2013b;a; 2010; Hinton et al., 2012; Hannun et al., 2014; Caruana et al., 2015; Alipanahi et al., 2015). However, in order to improve the safety of AI systems (Amodei et al., 2016) and avoid costly mistakes in high-risk applications, such as self-driving cars, it is desirable for models to yield estimates of uncertainty in their predictions. Ensemble methods are known to yield both improved predictive performance and robust uncertainty estimates (Gal & Ghahramani, 2016; Lakshminarayanan et al., 2017; Maddox et al., 2019). Importantly, ensemble approaches allow interpretable measures of uncertainty to be derived via a mathematically consistent probabilistic framework. Specifically, the overall *total uncertainty* can be decomposed into *data uncertainty*, or uncertainty due to inherent noise in the data, and *knowledge uncertainty*, which is due to the model having limited uncertainty of the test data (Malinin, 2019). Uncertainty estimates derived from ensembles have been applied to the detection of misclassifications, out-of-domain inputs and adversarial attack detection (Carlini & Wagner, 2017; Smith & Gal, 2018), and active learning (Kirsch et al., 2019). Unfortunately, ensemble methods may be computationally expensive to train and are always expensive during inference.

A class of models called *Prior Networks* (Malinin & Gales, 2018; 2019; Malinin, 2019; Sensoy et al., 2018) was proposed as an approach to modelling uncertainty in classification tasks by *emulating* an ensemble using a *single* model. Prior Networks parameterize a *higher order* conditional distribution over output distributions, such as the Dirichlet distribution. This enables Prior Networks to efficiently yield the same interpretable measures of *total*, *data* and *knowledge uncertainty* as an ensemble. Unlike ensembles, the behaviour of Prior Networks' higher-order distribution is specified via a loss function, such as reverse KL-divergence (Malinin & Gales, 2019), and training data. However, such Prior Networks yield predictive performance consistent with that of a single model trained via Maximum Likelihood, which is typically worse than that of an ensemble. This can be overcome via Ensemble Distribution Distillation (EnD$^2$) (Malinin et al., 2020), which is an approach that allows distilling an ensemble into Prior Network such that measures of ensemble diversity are preserved. This enables to retain both the predictive performance and uncertainty estimates of an ensemble at low computational and memory cost. Finally, it is important to point out that a related class of *evidential* methods has concurrently appeared (Sensoy et al., 2018; Amini et al., 2020). Structurally they yield models similar to Prior Networks, but are trained in a different fashion.

While Prior Networks have many attractive properties, they have only been applied to classification tasks. In this work we develop Prior Networks for *regression tasks* by considering the Normal-Wishart distribution - a higher-order distribution over the parameters of multivariate normal distributions. Specifically, we extend theoretical work from (Malinin, 2019), where such models are considered, but

never evaluated. We derive all measures of uncertainty, the reverse KL-divergence training objective, and the Ensemble Distribution Distillation objective in closed form. Regression Prior Networks are then evaluated on synthetic data, selected UCI datasets and the NYUv2 and KITTI monocular depth estimation tasks, where they are shown to yield comparable or better performance to state-of-the-art single-model and ensemble approaches. Crucially, they enable, via EnD$^2$, to retain the predictive performance and uncertainty estimates of an ensemble within a *single model*.

## 2 REGRESSION PRIOR NETWORKS

In this section we develop Prior Network models for regression tasks. While typical regression models yield point-estimate predictions, we consider *probabilistic regression models* which parameterizes a distribution $p(\boldsymbol{y}|\boldsymbol{x}, \boldsymbol{\theta})$ over the target $\boldsymbol{y} \in \mathcal{R}^K$. Typically, this is a normal distribution:

$$p(\boldsymbol{y}|\boldsymbol{x}, \boldsymbol{\theta}) = \mathcal{N}(\boldsymbol{y}|\boldsymbol{\mu}, \boldsymbol{\Lambda}), \quad \{\boldsymbol{\mu}, \boldsymbol{\Lambda}\} = \boldsymbol{f}(\boldsymbol{x}; \boldsymbol{\theta}) \tag{1}$$

where $\boldsymbol{\mu}$ is the mean, and $\boldsymbol{\Lambda}$ the precision matrix, a positive-definite symmetric matrix. While normal distributions are usually defined in terms of the covariance matrix $\boldsymbol{\Sigma} = \boldsymbol{\Lambda}^{-1}$, parameterization using the precision is more numerically stable during optimization (Bishop, 2006; Goodfellow et al., 2016). While a range of distributions over continuous random variables can be considered, we will consider the normal as it makes the least assumptions about the nature of $\boldsymbol{y}$ and is mathematically simple.

As in the case for classification, we can consider an ensemble of networks which parameterize multivariate normal distributions (MVN) $\{p(\boldsymbol{y}|\boldsymbol{x}, \boldsymbol{\theta}^{(m)})\}_{m=1}^M$. This ensemble can be interpreted as a set of draws from a higher-order implicit distribution over normal distributions. A Prior Network for regression would, therefore, emulate this ensemble by explicitly parameterizing a higher-order distribution over the parameters $\boldsymbol{\mu}$ and $\boldsymbol{\Lambda}$ of a normal distribution. One sensible choice is the formidable *Normal-Wishart distribution* (Murphy, 2012; Bishop, 2006), which is a conjugate prior to the MVN. This parallels how the Dirichlet distribution, the conjugate prior to the categorical, was used in classification Prior Networks. The Normal-Wishart distribution is defined as follows:

$$\mathcal{NW}(\boldsymbol{\mu}, \boldsymbol{\Lambda}|\boldsymbol{m}, \boldsymbol{L}, \kappa, \nu) = \mathcal{N}(\boldsymbol{\mu}|\boldsymbol{m}, \kappa\boldsymbol{\Lambda})\mathcal{W}(\boldsymbol{\Lambda}|\boldsymbol{L}, \nu) \tag{2}$$

where $\boldsymbol{m}$ and $\boldsymbol{L}$ are the *prior mean* and inverse of the positive-definite *prior scatter matrix*, while $\kappa$ and $\nu$ are the strengths of belief in each prior, respectively. The parameters $\kappa$ and $\nu$ are conceptually similar to *precision* of the Dirichlet distribution $\alpha_0$. The Normal-Wishart is a compound distribution which decomposes into a product of a conditional normal distribution over the mean and a Wishart distribution over the precision. Thus, a Regression Prior Network (RPN) parameterizes the Normal-Wishart distribution over the mean and precision of normal output distributions as follows:

$$p(\boldsymbol{\mu}, \boldsymbol{\Lambda}|\boldsymbol{x}, \boldsymbol{\theta}) = \mathcal{NW}(\boldsymbol{\mu}, \boldsymbol{\Lambda}|\boldsymbol{m}, \boldsymbol{L}, \kappa, \nu), \quad \{\boldsymbol{m}, \boldsymbol{L}, \kappa, \nu\} = \boldsymbol{\Omega} = \boldsymbol{f}(\boldsymbol{x}; \boldsymbol{\theta}) \tag{3}$$

where $\boldsymbol{\Omega} = \{\boldsymbol{m}, \boldsymbol{L}, \kappa, \nu\}$ is the set of parameters of the Normal-Wishart predicted by neural network. The posterior predictive of this model is the multivariate Student's $\mathcal{T}$ distribution (Murphy, 2012), which is the heavy-tailed generalization of the multivariate normal distribution:

$$p(\boldsymbol{y}|\boldsymbol{x}, \boldsymbol{\theta}) = \mathbb{E}_{p(\boldsymbol{\mu}, \boldsymbol{\Lambda}|\boldsymbol{x}, \boldsymbol{\theta})}[p(\boldsymbol{y}|\boldsymbol{\mu}, \boldsymbol{\Lambda})] = \mathcal{T}(\boldsymbol{y}|\boldsymbol{m}, \frac{\kappa + 1}{\kappa(\nu - K + 1)}\boldsymbol{L}^{-1}, \nu - K + 1) \tag{4}$$

In the limit, as $\nu \rightarrow \infty$, the $\mathcal{T}$ distribution converges to a normal distribution. The predictive posterior of the Prior Network given in equation 4 only has a defined mean and variance when $\nu > K + 1$.

Figure 1 depicts the desired behaviour of an ensemble of normal distributions sampled from a Normal-Wishart distribution. Specifically, the ensemble should be consistent for in-domain inputs in regions of low/high *data uncertainty*, as in figures 1a-b, and highly diverse both in the location of the mean and in the structure of the covariance for out-of-distribution inputs, as in figure 1c. Samples of continuous output distributions from a regression Prior Network should yield the same behaviour.

**Measures of Uncertainty** Given an RPN which displays these behaviours, we can compute closed-form expression for all uncertainty measures previously discussed for ensembles and Dirichlet Prior Networks (Malinin, 2019). We can obtain measures of *knowledge*, *total* and *data uncertainty* by considering the mutual information between $\boldsymbol{y}$ and the parameters of the output distribution $\{\boldsymbol{\mu}, \boldsymbol{\Lambda}\}$:

$$\underbrace{\mathcal{I}[\boldsymbol{y}, \{\boldsymbol{\mu}, \boldsymbol{\Lambda}\}]}_{\text{Knowledge Uncertainty}} = \underbrace{\mathcal{H}\big[\mathbb{E}_{p(\boldsymbol{\mu}, \boldsymbol{\Lambda}|\boldsymbol{x}, \boldsymbol{\theta})}[p(\boldsymbol{y}|\boldsymbol{\mu}, \boldsymbol{\Lambda})]\big]}_{\text{Total Uncertainty}} - \underbrace{\mathbb{E}_{p(\boldsymbol{\mu}, \boldsymbol{\Lambda}|\boldsymbol{x}, \boldsymbol{\theta})}\big[\mathcal{H}[p(\boldsymbol{y}|\boldsymbol{\mu}, \boldsymbol{\Lambda})]\big]}_{\text{Expected Data Uncertainty}} \tag{5}$$

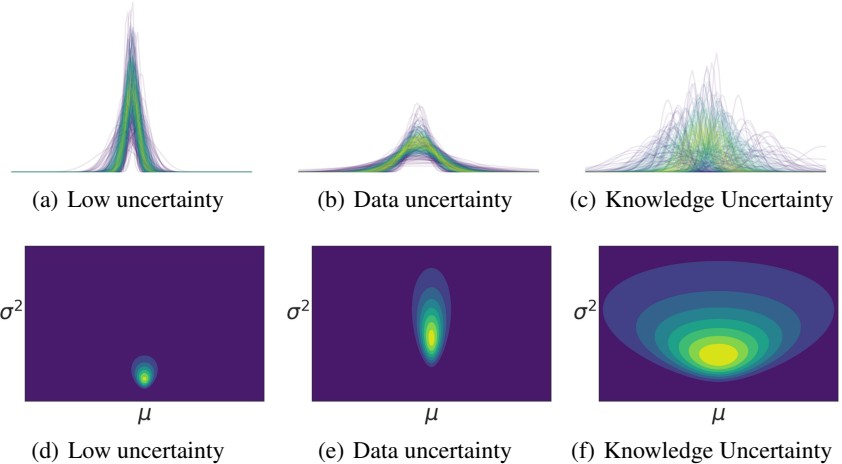

(a) Low uncertainty     (b) Data uncertainty     (c) Knowledge Uncertainty

(d) Low uncertainty     (e) Data uncertainty     (f) Knowledge Uncertainty

Figure 1: Desired behaviors of an ensemble of regression models. The bottom row displays the desired Normal-Wishart Distribution and the top row depicts Normal Distributions samples from it.

This expression consists of the difference between the differential entropy of the posterior predictive and the expected differential entropy of draws from the Normal-Wishart prior. We can also consider the *expected pairwise KL-divergence* (EPKL) between draws from the Normal-Wishart prior:

$$\mathcal{K}[\boldsymbol{y}, \{\boldsymbol{\mu}, \boldsymbol{\Lambda}\}] = -\mathbb{E}_{\mathrm{p}(\boldsymbol{y}|\boldsymbol{x},\boldsymbol{\theta})}[\mathbb{E}_{\mathrm{p}(\boldsymbol{\mu},\boldsymbol{\Lambda}|\boldsymbol{x},\boldsymbol{\theta})}[\ln \mathrm{p}(\boldsymbol{y}|\boldsymbol{\mu}, \boldsymbol{\Lambda})]] - \mathbb{E}_{\mathrm{p}(\boldsymbol{\mu},\boldsymbol{\Lambda}|\boldsymbol{x},\boldsymbol{\theta})}[\mathcal{H}[\mathrm{p}(\boldsymbol{y}|\boldsymbol{\mu}, \boldsymbol{\Lambda})]] \quad (6)$$

This is an upper bound on mutual information (Malinin, 2019). Notably, estimates of data uncertainty are unchanged. One practical use of EPKL is comparison with ensembles, as it is not possible to obtain a tractable expression for the mutual information of a regression ensemble (Malinin, 2019). Alternatively, we can consider measures of uncertainty derived via the law of total variance:

$$\underbrace{\mathbb{V}_{\mathrm{p}(\boldsymbol{\mu},\boldsymbol{\Lambda}|\boldsymbol{x},\boldsymbol{\theta})}[\boldsymbol{\mu}]}_{\text{Knowledge Uncertainty}} = \underbrace{\mathbb{V}_{\mathrm{p}(\boldsymbol{y}|\boldsymbol{x},\boldsymbol{\theta})}[\boldsymbol{y}]}_{\text{Total Uncertainty}} - \underbrace{\mathbb{E}_{\mathrm{p}(\boldsymbol{\mu},\boldsymbol{\Lambda}|\boldsymbol{x},\boldsymbol{\theta})}[\boldsymbol{\Lambda}^{-1}]}_{\text{Expected Data Uncertainty}} \quad (7)$$

This yields a similar decomposition to mutual information, but only first and second moments are considered. We provide closed-form expressions of (5)-(7) in appendix A. Note, however, that these variance-based measures are *not* scale-invariant, and are therefore sensitive to the scale of predictions which the model makes. This is relevant to applications such as depth-estimation, where we are likely to encounter images with a wide range of possible depths, and therefore varying scale of predictions. We omit the closed-form expressions for all terms here and instead provide them in appendix A.

**RKL training objective** Having discussed how to construct Prior Networks for regression, we now discuss how they can be trained. Prior Networks are trained using a multi-task loss, where an in-domain loss $\mathcal{L}_{in}$ and an out-of-distribution (OOD) loss $\mathcal{L}_{out}$ are jointly minimized:

$$\mathcal{L}(\boldsymbol{\theta}, \mathcal{D}_{tr}, \mathcal{D}_{out}) = \mathbb{E}_{\mathrm{p}_{tr}(\boldsymbol{y},\boldsymbol{x})}[\mathcal{L}_{in}(\boldsymbol{y}, \boldsymbol{x}, \boldsymbol{\theta})] + \gamma \cdot \mathbb{E}_{\mathrm{p}_{out}(\boldsymbol{x})}[\mathcal{L}_{out}(\boldsymbol{x}, \boldsymbol{\theta})] \quad (8)$$

The OOD loss is necessary to teach the model the limit of its knowledge (Malinin, 2019) and define a decision boundary between the in-domain and out-of-domain regions. In some applications the choice of OOD data can be particularly difficult. Normal distributions sampled from a Prior Network should be consistent and reflect the correct level of *data uncertainty* in-domain, and diverse in both mean and precision out-of-domain. Achieving the former is challenging, as the training data only consists of samples of inputs $\boldsymbol{x}$ and targets $\boldsymbol{y}$, there is no access to the underlying distribution, and associated data uncertainty, represented by the precision $\boldsymbol{\Lambda}$. Effectively, we are attempting to train a Normal-Wishart distribution from targets sampled from normal distribution that are sampled from the Normal-Wishart, rather than on the normal distribution samples themselves, which is challenging. However, it was shown that for Dirichlet Prior Networks minimizing the *reverse* KL-divergence between the model and an appropriate target Dirichlet *induces in expectation* the correct estimate of *data uncertainty* (Malinin & Gales, 2019). As the Normal-Wishart, like the Dirichlet, is a conjugate

prior and exponential family member, the precision can be *induced in expectation* by considering the *reverse* KL-divergence between the model $p(\boldsymbol{\mu}, \boldsymbol{\Lambda}|\boldsymbol{x}, \boldsymbol{\theta})$ and a target Normal-Wishart $p(\boldsymbol{\mu}, \boldsymbol{\Lambda}|\hat{\boldsymbol{\Omega}}^{(i)})$ corresponding to each $\boldsymbol{x}^{(i)}$. The appropriate Normal-Wishart is specified via Bayes's rule:

$$p(\boldsymbol{\mu}, \boldsymbol{\Lambda}|\hat{\boldsymbol{\Omega}}^{(i)}) \propto p(\boldsymbol{y}^{(i)}|\boldsymbol{\mu}, \boldsymbol{\Lambda})^{\hat{\beta}} p(\boldsymbol{\mu}, \boldsymbol{\Lambda}|\boldsymbol{\Omega}_0) \tag{9}$$

where $p(\boldsymbol{y}^{(i)}|\boldsymbol{\mu}, \boldsymbol{\Lambda})$ is a normal distribution and $\boldsymbol{\Omega}_0 = \{\boldsymbol{m}_0, \boldsymbol{L}_0, \kappa_0, \nu_0\}$ are the parameters of the prior $p(\boldsymbol{\mu}, \boldsymbol{\Lambda}|\boldsymbol{\Omega}_0)$ defined as follows:

$$\boldsymbol{m}_0 = \frac{1}{N} \sum_{i=1}^{N} \boldsymbol{y}^{(i)}, \ \boldsymbol{L}_0^{-1} = \frac{\nu_0}{N} \sum_{i=1}^{N} (\boldsymbol{y}^{(i)} - \boldsymbol{m}_0)(\boldsymbol{y}^{(i)} - \boldsymbol{m}_0)^{\mathsf{T}}, \ \kappa_0 = \epsilon, \ \nu_0 = K + 1 + \epsilon \tag{10}$$

In other words, we consider a *semi-informative prior* which corresponds to the mean and scatter matrix of marginal distribution $p(\boldsymbol{y})$, and we see each sample of the training data $\hat{\beta}$ times. The hyper-parameter $\hat{\beta}$ allows us to weigh the effect of the prior and the data. $\epsilon$ is a small value, like $10^{-2}$, so that $\kappa_0$ and $\nu_0$ yield a maximally un-informative, but proper predictive posterior. The reason to use a semi-informative prior is that in regression tasks, unlike classification tasks, uninformative priors are improper and lead to infinite differential entropy. Furthermore, we do know *something* about the data purely based on the marginal distribution, and it is sensible to use that as the prior. The reverse KL-divergence loss can then be expressed as:

$$\begin{aligned} \mathcal{L}(\boldsymbol{y}, \boldsymbol{x}, \boldsymbol{\theta}; \hat{\beta}, \boldsymbol{\Omega}_0) &= \text{KL}[p(\boldsymbol{\mu}, \boldsymbol{\Lambda}|\boldsymbol{x}, \boldsymbol{\theta})\|p(\boldsymbol{\mu}, \boldsymbol{\Lambda}|\hat{\boldsymbol{\Omega}}^{(i)})] \\ &= \hat{\beta} \cdot \mathbb{E}_{p(\boldsymbol{\mu}, \boldsymbol{\Lambda}|\boldsymbol{x}, \boldsymbol{\theta})} \big[ -\ln p(\boldsymbol{y}|\boldsymbol{\mu}, \boldsymbol{\Lambda}) \big] + \text{KL}[p(\boldsymbol{\mu}, \boldsymbol{\Lambda}|\boldsymbol{x}, \boldsymbol{\theta})\|p(\boldsymbol{\mu}, \boldsymbol{\Lambda}|\boldsymbol{\Omega}_0)] + Z \end{aligned} \tag{11}$$

where Z is a normalization constant independent of parameters $\boldsymbol{\theta}$. For in-domain data, $\hat{\beta}$ can be set to a large value, and for out-of-domain training data $\hat{\beta} = 0$, so that the model regresses to the prior. In-domain, the prior will add a degree of smoothing, which may prevent over-fitting and improve performance on small datasets. A large value of $\hat{\beta}$ means the model will yield a larger $\kappa$, $\nu$ for in-domain data. This will result in the first term of (11) being very close to the expected negative log-likelihood of the predictive posterior $p(\boldsymbol{y}|\boldsymbol{x}, \boldsymbol{\theta})$, thereby avoiding degradation of predictive performance. The derivation and closed-form expression for this loss is provided in appendix A.

**Ensemble Distribution Distillation** An exciting task which Prior Networks can solve is *Ensemble Distribution Distillation* (EnD$^2$) (Malinin et al., 2020), where the distribution of an ensemble's predictions is distilled into a single model. EnD$^2$ enables retaining an ensemble's improved predictive performance and uncertainty estimates within a single model at low cost. In contrast, standard Ensemble Distillation (EnD) minimizes the KL-divergence between a model and the ensemble:

$$\mathcal{L}_{\text{EnD}}(\boldsymbol{\phi}, \mathcal{D}_{trn}) = \frac{1}{NM} \sum_{i=1}^{N} \sum_{m=1}^{M} \text{KL}[p(\boldsymbol{y}|\boldsymbol{x}^{(i)}, \boldsymbol{\theta}^{(m)})\|p(\boldsymbol{y}|\boldsymbol{x}^{(i)}, \boldsymbol{\phi})]] \tag{12}$$

This loses information about ensemble diversity. A complication that occurs in probabilistic regression models is that the student model will try to fit a single normal distribution on top of a mixture distribution, spreading itself across each model. This may result in both poor predictive performance and poor estimates of uncertainty. An approach to both overcome this and retain information about diversity was considered in (Tran et al., 2020; Wu et al., 2020), where the student model parameterizes a mixture distribution, where each component of the mixture models a particular model in the ensemble. We refer to this as Mixture-Density Ensemble Distillation (MD-EnD):

$$p(\boldsymbol{y}|\boldsymbol{x}, \boldsymbol{\phi}) = \frac{1}{M} \sum_{m=1}^{M} \mathcal{N}(\boldsymbol{y}|\boldsymbol{\mu}^{(m)}, \boldsymbol{\Lambda}^{(m)}), \quad \{\boldsymbol{\mu}^{(m)}, \boldsymbol{\Lambda}^{(m)}\}_{m=1}^{M} = f(\boldsymbol{x}; \boldsymbol{\phi})$$

$$\mathcal{L}_{\text{MD-EnD}}(\boldsymbol{\phi}, \mathcal{D}_{trn}) = \frac{1}{NM} \sum_{i=1}^{N} \sum_{m=1}^{M} \text{KL}[p(\boldsymbol{y}|\boldsymbol{x}^{(i)}, \boldsymbol{\theta}^{(m)})\|\mathcal{N}(\boldsymbol{y}|\boldsymbol{\mu}^{(m)}, \boldsymbol{\Lambda}^{(m)}))]] \tag{13}$$

This clearly overcomes the issue of distributional mismatch, resolving any issues with poor predictive performance. It also allows the model to, theoretically, retain information about ensemble diversity. However, it may be challenging to fully replicate the behaviour of *each* ensemble member in detail

by having only multiple output heads - thus, splitting the model at an earlier point in the network may be necessary, which increases computational cost. EnD$^2$ avoids the problem by directly modelling the *bulk* behaviour of the ensemble, rather than the behavior of each individual model.

EnD$^2$ can be implemented for regression via RPNs as follows. Consider an ensemble $\{p(\boldsymbol{y}|\boldsymbol{x}, \boldsymbol{\theta}^{(m)})\}_{m=1}^{M}$, where each model yields the mean and precision of a normal distribution. We can define an empirical distribution over the mean and precision as follows:

$$\hat{p}(\boldsymbol{\mu}, \boldsymbol{\Lambda}, \boldsymbol{x}) = \left\{\{\boldsymbol{\mu}^{(mi)}, \boldsymbol{\Lambda}^{(mi)}\}_{m=1}^{M}, \boldsymbol{x}^{(i)}\right\}_{i=1}^{N} = \mathcal{D}_{trn} \tag{14}$$

EnD$^2$ can then be accomplished by minimizing the negative log-likelihood of the ensemble's mean and precision under the Normal-Wishart prior:

$$\mathcal{L}_{\mathrm{EnD^2}}(\boldsymbol{\phi}, \mathcal{D}_{trn}) = \mathbb{E}_{\hat{p}(\boldsymbol{\mu}, \boldsymbol{\Lambda}, \boldsymbol{x})}\left[-\ln p(\boldsymbol{\mu}, \boldsymbol{\Lambda}|\boldsymbol{x}; \boldsymbol{\phi})\right] = \mathbb{E}_{\hat{p}(\boldsymbol{x})}\left[\mathrm{KL}[\hat{p}(\boldsymbol{\mu}, \boldsymbol{\Lambda}|\boldsymbol{x})||p(\boldsymbol{\mu}, \boldsymbol{\Lambda}|\boldsymbol{x}; \boldsymbol{\phi})]\right] + Z \tag{15}$$

This is equivalent to minimizing the KL-divergence between the model and the empirical distribution of the ensemble. Note that here, unlike in the previous section, the parameters of a normal distribution are available for every input $\boldsymbol{x}$, making forward KL-divergence the appropriate loss function. However, while this is a theoretically sound approach, the optimization might be numerically challenging. Similarly to (Malinin et al., 2020) we propose a temperature-annealing trick to make the optimization process easier. First, the ensemble is reduced to it's mean:

$$\boldsymbol{\mu}_T^{(mi)} = \frac{2}{T+1}\boldsymbol{\mu}^{(mi)} + \frac{T-1}{T+1}\bar{\boldsymbol{\mu}}^{(i)}, \quad \bar{\boldsymbol{\mu}}^{(i)} = \frac{1}{M}\sum_{m=1}^{M}\boldsymbol{\mu}^{(mi)}$$

$$\boldsymbol{\Lambda}_T^{-1(mi)} = \frac{2}{T+1}\boldsymbol{\Lambda}^{-1(mi)} + \frac{T-1}{T+1}\bar{\boldsymbol{\Lambda}}^{-1(i)}, \quad \bar{\boldsymbol{\Lambda}}^{-1(i)} = \frac{1}{M}\sum_{m=1}^{M}\bar{\boldsymbol{\Lambda}}^{-1(mi)} \tag{16}$$

We use inverses of the precision matrix $\boldsymbol{\Lambda}$ because we are interpolating the covariance matrices $\boldsymbol{\Sigma}$. Secondly, the predicted $\tilde{\kappa} = T\kappa$ and $\tilde{\nu} = T\nu$ are multiplied by $T$ in order to make the Normal-Wishart sharp around the mean. The loss is *divided* by T to avoid scaling the gradients by T, yielding:

$$p_T(\boldsymbol{\mu}, \boldsymbol{\Lambda}|\boldsymbol{x}, \boldsymbol{\phi}) = \mathcal{N}\mathcal{W}(\boldsymbol{\mu}, \boldsymbol{\Lambda}|\boldsymbol{m}, \boldsymbol{L}, T\kappa, T\nu), \quad \{\boldsymbol{m}, \boldsymbol{L}, \kappa, \nu\} = \boldsymbol{f}(\boldsymbol{x}; \boldsymbol{\phi})$$

$$\mathcal{L}_{\mathrm{EnD^2}}(\boldsymbol{\phi}, \mathcal{D}_{trn}; T) = \frac{1}{T}\mathbb{E}_{\hat{p}_T(\boldsymbol{\mu}, \boldsymbol{\Lambda}, \boldsymbol{x})}\left[-\ln p_T(\boldsymbol{\mu}, \boldsymbol{\Lambda}|\boldsymbol{x}; T, \boldsymbol{\phi})\right] \tag{17}$$

This splits learning into two phases. First, when the temperature is high, the model learns to match the ensemble's mean (first moment). Second, as the temperature is annealed down to 1, the model will gradually focus on learning higher moments of the ensemble's distribution. This trick may be necessary, as the ensemble may have a highly non-Normal-Wishart distribution, which may be challenging to learn. Note that for EnD$^2$ it may be better to parameterize the *Normal-inverse-Wishart* distribution over the mean and covariance due to numerical stability concerns. However, for consistency, we describe EnD$^2$ in terms of the Normal-Wishart. Finally, we emphasize that EnD$^2$ does not require OOD training data, unlike the RKL objective above. This eliminates the non-trivial challenge of finding appropriate OOD data.

**Related Approaches** It is necessary to mention evidential approaches, specifically Deep Evidential Regression (Amini et al., 2020). Structurally, it considers models which are similar to RPNs, and is thus compared to in section 5. However, they do not use OOD training data nor attempt to emulate and generalize an ensemble's behaviour to enforce high-uncertainty behaviour in OOD regions. Rather, they are trained by maximizing the likelihood of the $\mathcal{T}$ distribution (equation (4)) together with an *evidence regularizer* which forces the model to yield high $\nu$, $\kappa$ on regions with low absolute error, and low evidence in regions of high L1 error. However, this seems susceptible to pathologies, such as making the model overconfident if it over-fits the training data and achieves zero MAE, or introducing an inverse correlation between $\nu$, $\kappa$ and the scale of predictions in datasets where the scale of predictions range widely. Thus, while they yield encouraging results, their principle of action remains unclear. Another related recent research direction is in developing computationally efficient ensembles (Wen et al., 2020). Here, rather than emulating an ensemble using a single model, the goal is to generate high diversity in predictions while re-using large portions of a neural network model, thereby making for compact ensembles. While such approaches are more computationally efficient than using multiple independent models and use only a little more memory than a single model *on disk*, they still require using M times as much GPU memory *at run time*, which can be a significant limitation in resource constrained applications, like mobile devices or autonomous vehicles.

# 3 EXPERIMENTS ON SYNTHETIC DATA

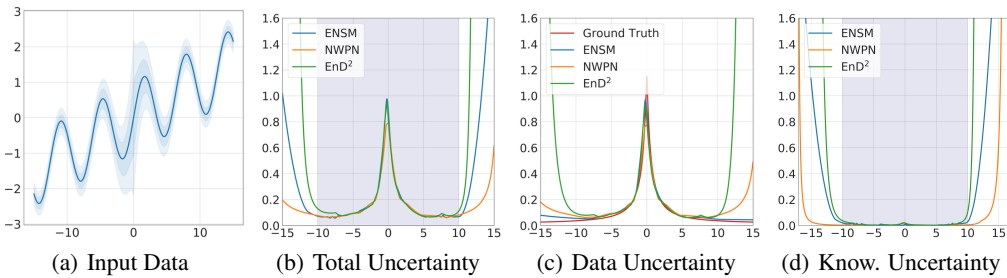

| (a) Input Data | (b) Total Uncertainty | (c) Data Uncertainty | (d) Know. Uncertainty |

Figure 2: Comparison of different models on synthetic data $y \sim \mathcal{N}(\sin x + \frac{x}{10}, \frac{1}{|x|+1} + 0.01)$. Gray area indicates training data region.

We first examine Regression Prior Networks on a synthetic one-dimensional dataset with additive heteroscedastic noise. We compare Regression Prior Network trained via RKL (NWPN) and distribution-distillation of an ensemble (EnD$^2$) to Deep Ensembles (ENSM). The ensemble consists of 10 models that yield a Normal output distribution and are trained via maximum likelihood. All models use the same 2-layer architecture with 30 ReLU units. Details of the setup are available in appendix B. The NWPN was trained via reverse KL-divergence (11), and Ensemble Distribution Distillation using the loss in equation (17). In-domain training data for all models is sampled between [-10,10], and OOD training data for the NWPN model is sampled from $[-25, 20] \cup [20, 25]$. Measures of *total*, *data* and *knowledge uncertainty* are obtained via the law of total variance (7).

The results presented in Figure 2 show several trends. Firstly, the total uncertainty of all models is high in the region of high heteroscedastic noise as well as out-of-domain. Secondly, *total uncertainty* decomposes into *data uncertainty* and *knowledge uncertainty*. The former is high in the region of high heteroscedastic noise and has undefined behavior out-of-domain, while the latter is low in-domain and large out-of-distribution. Third, EnD$^2$ successfully replicates the ensemble's estimates of uncertainty, though they are consistently larger, especially estimates of *data uncertainty* out-of-domain. This is a consequence of the ensemble being non-Normal-Wishart distributed when it is diverse, leading the EnD$^2$ Prior Network to over-estimate support. Thus, these results validate the principle claims that Regression Prior Networks can emulate an ensemble's behavior via multi-task training using the RKL objective or via EnD$^2$ and that they yield interpretable measures of uncertainty.

# 4 EXPERIMENTS ON UCI DATA

In this section, we evaluate Normal-Wishart Prior Networks trained via reverse-KL divergence (11) (NWPN) and Ensemble Distribution Distillation (EnD$^2$) relative to a Deep-Ensemble (ENSM) baseline on selected UCI datasets. Other ensemble-methods are not considered, as Deep Ensembles have been shown to consistently outperform them using fewer ensemble members (Ashukha et al., 2020; Ovadia et al., 2019; Fort et al., 2019). We follow the experimental setup of (Lakshminarayanan et al., 2017) with several changes, detailed in appendix C. Out-of-distribution training data for NWPN is generated using a factor analysis model. This model is a linear generative model that learns to approximate in-domain data with $\hat{x} \sim \mathcal{N}(\mu, WW^T + \Psi)$, where $[W, \mu, \Psi]$ are model parameters. The out-of-domain training examples are then sampled from $\mathcal{N}(\mu, 3WW^T + 3\Psi)$, such that they are further from the in-domain region. Table 1 shows a comparison of all models in terms of NLL and RMSE. Single is a Gaussian model, ENSM is an ensemble of Gaussian models. Unsurprisingly, ensembles yield the best RMSE, though both NWPN and EnD$^2$ generally give comparable NLL scores. Furthermore, EnD$^2$ comes close to or matches the performance of the ensemble and outperforms NWPN.

In Table 2 we compare uncertainty measures derived from all models on the tasks of error detection and OOD detection. To evaluate error detection a Prediction Rejection Ratio (PRR) is used. It shows what part of the best possible error-detection performance our algorithm covers and is defined in appendix C. For the evaluation of OOD-detection performance, we took parts of other UCI datasets as

Table 1: RMSE and NLL of models on UCI datasets. Datasets listed in order of increasing size. Results on remaining UCI datasets are available in appendix C

| Data | RMSE ($\downarrow$) | | | | NLL ($\downarrow$) | | | |
|------|--------|------|--------|------|--------|------|--------|------|
| | Single | ENSM | EnD$^2$ | NWPN | Single | ENSM | EnD$^2$ | NWPN |
| wine | $0.65 \pm 0.01$ | $\mathbf{0.63 \pm 0.01}$ | $\mathbf{0.63 \pm 0.01}$ | $\mathbf{0.63 \pm 0.01}$ | $1.24 \pm 0.16$ | $0.96 \pm 0.03$ | $\mathbf{0.91 \pm 0.02}$ | $0.93 \pm 0.02$ |
| power | $4.07 \pm 0.07$ | $\mathbf{4.00 \pm 0.07}$ | $4.06 \pm 0.07$ | $4.09 \pm 0.07$ | $2.82 \pm 0.02$ | $\mathbf{2.79 \pm 0.02}$ | $\mathbf{2.79 \pm 0.01}$ | $2.81 \pm 0.01$ |
| MSD | $9.08 \pm 0.00$ | $\mathbf{8.92 \pm 0.00}$ | $8.94 \pm 0.00$ | $9.07 \pm 0.00$ | $3.51 \pm 0.00$ | $\mathbf{3.39 \pm 0.00}$ | $\mathbf{3.39 \pm 0.00}$ | $3.41 \pm 0.00$ |

OOD data. We made sure that the OOD-data comes from different domains and feature distributions are different. Columns of each OOD dataset are normalized using statistics derived from the in-domain training dataset. Details are available in appendix C. The results show that all models achieve comparable error-detection using measures of *total uncertainty*. In terms of OOD detection, EnD$^2$ generally reproduces the ensemble's behavior, while NWPN usually performs worse. However, on the MSD dataset, NWPN yields the best performance. This may be due to the nature of the OOD training data - it may simply better suited to MSD OOD detection. Furthermore, the UCI datasets are generally small and have low input dimensionality - MSD, the largest, has 95 features. Therefore, it is difficult to assess the superiority of any particular model on these simple datasets - all we can say that they generally perform comparably. In the next section, we validate Regression Prior Networks on a more complex, larger-scale task.

Table 2: PRR and OOD detection scores

| Data | Model | PRR ($\uparrow$) | | AUC-ROC ($\uparrow$) | | |
|------|-------|------|------|------|------|------|
| | | $\mathcal{H}[\mathbb{E}]$ | $\mathbb{V}[\boldsymbol{y}]$ | $\mathcal{I}$ | $\mathcal{K}$ | $\mathbb{V}[\boldsymbol{\mu}]$ |
| wine | ENSM | - | $\mathbf{0.32 \pm 0.02}$ | - | $0.58 \pm 0.01$ | $0.56 \pm 0.02$ |
| | EnD$^2$ | $0.30 \pm 0.02$ | $0.30 \pm 0.02$ | $\mathbf{0.65 \pm 0.01}$ | $\mathbf{0.65 \pm 0.01}$ | $\mathbf{0.65 \pm 0.03}$ |
| | NWPN | $0.30 \pm 0.03$ | $0.30 \pm 0.03$ | $0.64 \pm 0.01$ | $0.64 \pm 0.01$ | $0.53 \pm 0.02$ |
| power | ENSM | - | $\mathbf{0.23 \pm 0.01}$ | - | $0.64 \pm 0.02$ | $0.62 \pm 0.02$ |
| | EnD$^2$ | $\mathbf{0.23 \pm 0.01}$ | $\mathbf{0.23 \pm 0.01}$ | $\mathbf{0.66 \pm 0.01}$ | $\mathbf{0.66 \pm 0.01}$ | $0.50 \pm 0.02$ |
| | NWPN | $0.20 \pm 0.02$ | $0.20 \pm 0.02$ | $0.56 \pm 0.01$ | $0.56 \pm 0.01$ | $0.31 \pm 0.02$ |
| msd | ENSM | - | $\mathbf{0.64 \pm 0.00}$ | - | $0.55 \pm 0.0$ | $0.62 \pm 0.0$ |
| | EnD$^2$ | $0.63 \pm 0.00$ | $0.63 \pm 0.00$ | $0.50 \pm 0.0$ | $0.50 \pm 0.0$ | $0.65 \pm 0.0$ |
| | NWPN | $\mathbf{0.64 \pm 0.00}$ | $\mathbf{0.64 \pm 0.00}$ | $0.73 \pm 0.0$ | $0.73 \pm 0.0$ | $\mathbf{0.75 \pm 0.0}$ |

## 5 MONOCULAR DEPTH ESTIMATION EXPERIMENTS

Having established that the proposed methods work on par with or better than ensemble methods on the UCI datasets, we now examine them on the large-scale NYU Depth v2 (Nathan Silberman & Fergus, 2012) and KITTI (Menze & Geiger, 2015) depth-estimation tasks. In this section, the base model is DenseDepth (DD), which defines a U-Net like architecture on top of DenseNet-169 features (Alhashim & Wonka, 2018). The original approach trains it on inverted targets using a combination of L1, SSIM, and Image-Gradient losses. We replace this with NLL training using a Gaussian model, which yields mean and precision for each pixel (Single). We also use original targets from the dataset. The rest of the data pre-processing, augmentation, optimization, and evaluation protocol is kept unchanged. On the challenging KITTI benchmark (Geiger et al., 2013) all models are evaluated on the split proposed by (Eigen et al., 2014).

We consider the following baselines: a single Gaussian model (Single), Deep-Ensemble of 5 Gaussian models (ENSM), ensemble distillation (EnD) (Hinton et al., 2015), mixture density ensemble distillation (MD-EnD) Tran et al. (2020); Wu et al. (2020) and Deep Evidential Regression (Amini et al., 2020). We distribution-distill the ensemble into a Regression Prior Network (EnD$^2$) with a per-pixel Normal-Wishart distribution. We also examine training an RPN with the RKL loss function (NWPN). On the NYU dataset, we take KITTI as OOD training data, and on KITTI, we use NYU as OOD training data. We retrain all models 4 times with different random seeds and report mean. Distribution-distillation is done with temperature annealing ($T = 1.0$ vs. $T = 10.0$). For $T = 10.0$ we train with initial temperature for $20\%$ of epochs, linearly decay it to $1.0$ during $60\%$ of epochs, fine-tune with $T = 1.0$ for remaining epochs. We found that this greatly stabilized training.

We report standard predictive performance metrics for depth estimation (Eigen et al., 2014), a detailed description can be found in appendix D. Results in table 3 show that all probabilistic models either

outperform or work on par with the original DenseDepth on both datasets. $EnD^2$ and MD-EnD achieve performance closest to the ensemble, with $End^2$ doing marginally better than MD-EnD on Kitti. At the same time both DER and NWPN achieve performance comparable to a single model, though NWPN does marginally worse on NYU. In terms of test-set negative log-likelihood, a metric of calibration, DER, NWPN and $EnD^2$ outperform all other approaches, including the ensemble. With regards to $EnD^2$ this may be due to the ensemble being poorly modeled by a Normal-Wishart distribution, leading the Prior Network to overestimate the distribution's support, and therefore yield less overconfident prediction. NWPN and DER may simply be well-regularized and smoothed. Finally, both EnD and MD-EnD are significantly worse in terms of calibration (NLL) on both datasets. EnD also achieves both poor predictive and calibration performance, which is likely an artifact of trying to model a mixture of Gaussians with a single Gaussian.

Table 3: Predictive Performance comparison

| Method | NYUv2 Predictive Performance | | | | | | | KITTI Predictive Performance | | | | | | |
|---|---|---|---|---|---|---|---|---|---|---|---|---|---|---|
| | $\delta_1(\uparrow)$ | $\delta_2(\uparrow)$ | $\delta_3(\uparrow)$ | rel($\downarrow$) | rmse($\downarrow$) | $\log_{10}(\downarrow)$ | NLL($\downarrow$) | $\delta_1(\uparrow)$ | $\delta_2(\uparrow)$ | $\delta_3(\uparrow)$ | rel($\downarrow$) | rmse($\downarrow$) | $\log_{10}(\downarrow)$ | NLL($\downarrow$) |
| DD | 0.847 | 0.972 | 0.993 | 0.124 | 0.468 | 0.054 | - | 0.886 | 0.965 | 0.986 | 0.093 | 4.170 | - | - |
| ENSM 5 | **0.862** | **0.975** | **0.994** | **0.117** | **0.438** | **0.051** | 0.76 | **0.932** | **0.989** | **0.998** | **0.073** | **3.355** | **0.032** | 1.94 |
| Single | 0.852 | 0.971 | 0.993 | 0.122 | 0.456 | 0.053 | 5.74 | 0.924 | 0.987 | 0.997 | 0.078 | 3.545 | 0.034 | 2.98 |
| NWPN | 0.842 | 0.968 | 0.992 | 0.126 | 0.472 | 0.055 | **-1.60** | 0.920 | 0.986 | 0.997 | 0.077 | 3.525 | 0.035 | 1.52 |
| EnD | 0.851 | 0.971 | 0.993 | 0.122 | 0.458 | 0.053 | 9.11 | 0.915 | 0.984 | 0.997 | 0.079 | 3.936 | 0.036 | 3.27 |
| MD-EnD | 0.858 | 0.972 | 0.992 | 0.121 | 0.451 | 0.051 | 8.48 | 0.925 | 0.987 | 0.997 | 0.079 | 3.446 | 0.034 | 2.30 |
| $EnD^2$ | 0.855 | 0.972 | 0.993 | 0.120 | 0.451 | 0.052 | -1.47 | 0.928 | 0.988 | 0.998 | 0.075 | 3.367 | 0.033 | **1.42** |
| DER | 0.847 | 0.969 | 0.992 | 0.125 | 0.464 | 0.053 | -1.04 | 0.926 | 0.986 | 0.997 | 0.078 | 3.552 | 0.034 | 1.71 |

In table 4 we assess all models on the task of out-of-domain input detection. Two OOD test-datasets are considered: LSUN-church (LSN-C) and LSUN-bed (LSN-B) (Yu et al., 2015), which consist of images of churches and bedrooms. The latter is most similar to NYU Depth-V2 and more challenging to detect. OOD images are center-cropped and re-scaled to the in-domain data. With regards to KITTI, which consists of outdoor images of roads and displays a large range of depth in each image, the OOD data is far closer to the camera than the in-domain data as a result of a crop-and-scale prepossessing. Examples of this a provided in appendix D.4.

Results show several trends. $EnD^2$ consistently outperforms the original ensemble using measures of *knowledge uncertainty* ($\mathcal{I}$, $\mathcal{K}$ and $\mathbb{V}[\boldsymbol{\mu}]$). However, when considering measures of *total uncertainty* ($\mathcal{H}[\mathbb{E}]$, $\mathbb{V}[\boldsymbol{y}]$), the ensemble tends to yield superior performance on NYU. This is likely due to the over-estimation of the support of the ensemble's distribution by $EnD^2$. In contrast, EnD and MD-EnD perform worse than a single model, likely due to either failing to match the ensemble due to distributional mismatch (EnD), or having limited capacity to model the individual behaviour of each ensemble member (MD-EnD). In contrast, $EnD^2$ models the 'bulk' behaviour of the ensemble, and does not suffer from either issue, which highlights its importance.

At the same time, while NWPN yields the best OOD-detection performance on KITTI, it fails to be robust on NYU. It is necessary to point out that we found training RPNs with RKL challenging in this setting, as it is non-trivial to define what OOD is, especially for depth estimation. In this paper, we consider a different dataset to be OOD. An ablation study with varying OOD weight (appendix D) shows a trade-off between predictive quality and OOD detection quality. Appropriate training yields the best OOD detection performance (KITTI), and if the balance is incorrect, the performance is poor (NYU). We speculate that this is because depth estimation is a task sensitive to local features, while discriminating between datasets requires global features, so the two tasks interfere. This highlights the value of $EnD^2$, which does *not* require OOD training data or additional hyperparameters, and yields good predictive and OOD detection performance.

Interestingly, variance-based measures outperform information-theoretic measures of *total uncertainty*, and sometimes *knowledge uncertainty* on NYU, but fail on KITTI. This is a result of their sensitivity to scale. The models predict very low depth values for OOD data, which means that they have lower entropy and variance. In contrast, MI and EPKL, which are scale-invariant, are not affected. The issue of scale sensitivity is also the reason why Deep Evidential Regression (DER) completely fails on KITTI. As discussed in section 2, the evidence regularizer forces the model to yield low uncertainty measures in regions of low error. Therefore, when it observes data too close to the camera, DER tends to detect it as in-domain. Additional OOD detection results presented in appendix D.4.

Table 4: OOD detection % AUC-ROC ($\uparrow$) comparison

| Method | OOD | NYUv2 vs LSUN OOD Detection | | | | | KITTI vs LSUN OOD Detection | | | | |
|--------|-----|------------------|----------------|----------------|----------------|----------------|------------------|----------------|----------------|----------------|----------------|
| | | $\mathcal{H}[\mathbb{E}]$ | $\mathbb{V}[\boldsymbol{y}]$ | $\mathcal{I}$ | $\mathcal{K}$ | $\mathbb{V}[\boldsymbol{\mu}]$ | $\mathcal{H}[\mathbb{E}]$ | $\mathbb{V}[\boldsymbol{y}]$ | $\mathcal{I}$ | $\mathcal{K}$ | $\mathbb{V}[\boldsymbol{\mu}]$ |
| Single | | 0.69 | 0.69 | - | - | - | 0.02 | 0.02 | - | - | - |
| NWPN | | 0.801 | 0.801 | 0.199 | 0.199 | 0.799 | 0.999 | 0.999 | **1.0** | **1.0** | **1.0** |
| EnD | | 0.646 | 0.646 | - | - | - | 0.003 | 0.003 | - | - | - |
| EnD$^2$ | LSN-B | 0.724 | 0.733 | **0.817** | 0.806 | 0.770 | 0.015 | 0.017 | 0.887 | 0.868 | 0.040 |
| DER | | 0.722 | 0.732 | 0.717 | 0.638 | 0.728 | 0.028 | 0.030 | 0.029 | 0.005 | 0.035 |
| MD-EnD | | - | 0.630 | - | 0.488 | 0.502 | - | 0.004 | - | 0.448 | 0.033 |
| ENSM | | - | 0.723 | - | 0.672 | 0.745 | - | 0.032 | - | 0.822 | 0.097 |
| Single | | 0.845 | 0.845 | - | - | - | 0.023 | 0.023 | - | - | - |
| NWPN | | **0.993** | **0.993** | 0.003 | 0.003 | 0.992 | 0.994 | 0.994 | **1.0** | **1.0** | 0.998 |
| EnD | | 0.703 | 0.703 | - | - | - | 0.004 | 0.004 | - | - | - |
| EnD$^2$ | LSN-C | 0.882 | 0.893 | 0.964 | 0.952 | 0.928 | 0.018 | 0.020 | 0.834 | 0.806 | 0.036 |
| DER | | 0.877 | 0.877 | 0.791 | 0.675 | 0.878 | 0.034 | 0.035 | 0.035 | 0.009 | 0.040 |
| MD-EnD | | - | 0.698 | - | 0.119 | 0.422 | - | 0.012 | - | 0.506 | 0.031 |
| ENSM | | - | 0.887 | - | 0.696 | 0.886 | - | 0.036 | - | 0.779 | 0.098 |

Figure 3: Comparison of uncertainty measures between ensembles and EnD$^2$. For two input images, we demonstrate the difference between prediction and ground truth (Error), measures of Total Variance (Total) and EPKL (Knowledge) obtained from ensembles and our model. The left and right images corresponds to models trained on the KITTI an NYU datasets, respectively.

Last, figure 3 shows the error and estimates of *total* and *knowledge uncertainty* of the ensemble and an EnD$^2$ model for the same input image. Both the ensemble and our model effectively decompose uncertainty. *Total uncertainty* is correlated with an error and is large at object boundaries and distant points while *knowledge uncertainty* concentrates on the interior of unusual objects. EnD$^2$ yields both error and uncertainty measures, which are very similar to that of the original ensemble. This demonstrates that EnD$^2$ can emulate not only the predictive performance of the ensemble but also the behavior of the ensemble's measures of uncertainty. Further comparisons are provided in appendix D.

## 6 CONCLUSION

This work proposed Regression Prior Networks, yielding a set of general, efficient, and interpretable uncertainty estimation approaches for regression. A Regression Prior Network (RPN) predicts the parameters of a Normal-Wishart distribution, enabling it to efficiently represent ensembles of regression models, allowing interpretable measures of uncertainty to be obtained at a low computational cost. In this work closed-form measures of *total*, *data* and *knowledge uncertainty* are obtained for Normal-Wishart RPNs. Two RPN training approaches are proposed. First, the reverse-KL divergence between the model and a target Normal-Wishart distribution is described, allowing the behaviour of an RPN to be explicitly controlled but requiring an OOD training dataset. Second, Ensemble Distribution Distillation (EnD$^2$) is used, where an ensemble of regression models is distilled into an RPN such that it retains the improved predictive performance and uncertainty estimates of the original ensemble. This approach is particularly useful when it is challenging to define an appropriate out-of-domain training dataset, such as in depth-estimation. The properties of RPNs were evaluated on selected UCI datasets and two large-scale monocular depth-estimation tasks. Here Ensemble Distribution Distilled RPNs, which do *not* need OOD training data, were shown to outperform other single-model and distillation approaches in terms of predictive performance, and all models in overall OOD-detection quality. This demonstrates its value as a computationally cheap, general-purpose uncertainty estimation approach for regressions tasks.

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

# A   DERIVATIONS FOR NORMAL-WISHART PRIOR NETWORKS

The current appendix provides mathematical details of the Normal-Wishart distribution and derivations of the reverse-KL divergence loss, ensemble distribution distillation and all uncertainty measures.

## A.1   NORMAL-WISHART DISTRIBUTION

The Normal-Wishart distribution is a conjugate prior over the mean $\boldsymbol{\mu}$ and precision $\boldsymbol{\Lambda}$ of a normal distribution, defined as follows

$$\mathrm{p}(\boldsymbol{\mu}, \boldsymbol{\Lambda} | \boldsymbol{\Omega}) = \mathcal{NW}(\boldsymbol{\mu}, \boldsymbol{\Lambda} | \boldsymbol{m}, \boldsymbol{L}, \kappa, \nu) = \mathcal{N}(\boldsymbol{\mu} | \boldsymbol{m}, \kappa\boldsymbol{\Lambda})\mathcal{W}(\boldsymbol{\Lambda} | \boldsymbol{L}, \nu); \tag{18}$$

where $\boldsymbol{\Omega} = \{\boldsymbol{m}, \boldsymbol{L}, \kappa, \nu\}$ are the parameters predicted by neural network, $\mathcal{N}$ is the density of the Normal and $\mathcal{W}$ is the density of the Wishart distribution. Here, $\boldsymbol{m}$ and $\boldsymbol{L}$ are the *prior mean* and inverse of the positive-definite *prior scatter matrix*, while $\kappa$ and $\nu$ are the strengths of belief in each prior, respectively. The parameters $\kappa$ and $\nu$ are conceptually similar to *precision* of the Dirichlet distribution $\alpha_0$. The Normal-Wishart is a compound distribution which decomposes into a product of a conditional normal distribution over the mean and an Wishart distribution over the precision:

$$\mathcal{N}(\boldsymbol{\mu} | \boldsymbol{m}, \kappa\boldsymbol{\Lambda}) = \frac{\kappa^{\frac{D}{2}} |\boldsymbol{\Lambda}|^{\frac{1}{2}}}{2\pi^{\frac{D}{2}}} \exp\left( -\frac{\kappa}{2}(\boldsymbol{\mu} - \boldsymbol{m})^{\mathrm{T}}\boldsymbol{\Lambda}(\boldsymbol{\mu} - \boldsymbol{m}) \right)$$
$$\mathcal{W}(\boldsymbol{\Lambda} | \boldsymbol{L}, \nu) = \frac{|\boldsymbol{\Lambda}|^{\frac{\nu - K - 1}{2}} \exp(-\frac{1}{2}\mathrm{Tr}(\boldsymbol{\Lambda}\boldsymbol{L}^{-1}))}{2^{\frac{\nu K}{2}} \Gamma_K(\frac{\nu}{2})|\boldsymbol{L}|^{\frac{\nu}{2}}}; \ \boldsymbol{\Lambda}, \boldsymbol{L} \succ 0, \nu > K - 1. \tag{19}$$

where $\Gamma_K(\cdot)$ is the *multivariate gamma function* and $K$ is the dimensionality of $\boldsymbol{y}$. From (Murphy, 2012) the posterior predictive of this model is the multivariate T-distribution:

$$\mathrm{p}(\boldsymbol{y} | \boldsymbol{x}, \boldsymbol{\theta}) = \mathbb{E}_{\mathrm{p}(\boldsymbol{\mu}, \boldsymbol{\Lambda} | \boldsymbol{x}, \boldsymbol{\theta})}\big[\mathrm{p}(\boldsymbol{y} | \boldsymbol{\mu}, \boldsymbol{\Lambda})\big] = \mathcal{T}(\boldsymbol{y} | \boldsymbol{m}, \frac{\kappa + 1}{\kappa(\nu - K + 1)}\boldsymbol{L}^{-1}, \nu - K + 1). \tag{20}$$

The $\mathcal{T}$ distribution is heavy-tailed generalization of the multivariate normal distribution defined as:

$$\mathcal{T}(\boldsymbol{y} | \boldsymbol{\mu}, \boldsymbol{\Sigma}, \nu) = \frac{\Gamma(\frac{\nu + K}{2})}{\Gamma(\frac{\nu}{2})\nu^{\frac{K}{2}}\pi^{\frac{K}{2}}|\boldsymbol{\Sigma}|^{\frac{1}{2}}} \left( 1 + \frac{1}{\nu}(\boldsymbol{y} - \boldsymbol{\mu})^{\mathrm{T}}\boldsymbol{\Sigma}^{-1}(\boldsymbol{y} - \boldsymbol{\mu}) \right)^{-\frac{(\nu + K)}{2}}, \ \nu \geq 0; \tag{21}$$

where $\nu$ is the number of degrees of freedom. However, the mean is only defined when $\nu > 1$ and the variance is defined only when $\nu > 2$.

## A.2   REVERSE KL-DIVERGENCE TRAINING OBJECTIVE

Now let us consider in greater detail the reverse KL-divergence training objective (11):

$$\mathcal{L}(\boldsymbol{y}, \boldsymbol{x}, \boldsymbol{\theta}; \hat{\beta}, \boldsymbol{\Omega}_0) = \hat{\beta} \cdot \mathbb{E}_{\mathrm{p}(\boldsymbol{\mu}, \boldsymbol{\Lambda} | \boldsymbol{x}, \boldsymbol{\theta})}\big[-\ln \mathrm{p}(\boldsymbol{y} | \boldsymbol{\mu}, \boldsymbol{\Lambda})\big] + \mathrm{KL}[\mathrm{p}(\boldsymbol{\mu}, \boldsymbol{\Lambda} | \boldsymbol{x}, \boldsymbol{\theta}) \| \mathrm{p}(\boldsymbol{\mu}, \boldsymbol{\Lambda} | \boldsymbol{\Omega}_0)] + Z \tag{22}$$

where $\boldsymbol{\Omega}_0 = [\boldsymbol{m}_0, \boldsymbol{L}_0, \kappa_0, \nu_0]$ are prior parameters that we set manually as discussed in section 2. It is necessary to show why the reverse KL-divergence objective will yield the correct level of data uncertainty. Lets consider taking the expectation of the first term in (11) with respect to the *true distribution* of targets $\mathrm{p}_{tr}(\boldsymbol{y} | \boldsymbol{x})$. Trivially, we can show that by exchanging the order of expectation, that we are optimizing the expected cross-entropy between samples from the Normal-Wishart and the true distribution:

$$\mathbb{E}_{\mathrm{p}_{tr}(\boldsymbol{y} | \boldsymbol{x})}\big[\mathbb{E}_{\mathrm{p}(\boldsymbol{\mu}, \boldsymbol{\Lambda} | \boldsymbol{x}, \boldsymbol{\theta})}\big[-\ln \mathrm{p}(\boldsymbol{y} | \boldsymbol{\mu}, \boldsymbol{\Lambda})\big]\big] = \mathbb{E}_{\mathrm{p}(\boldsymbol{\mu}, \boldsymbol{\Lambda} | \boldsymbol{x}, \boldsymbol{\theta})}\big[\mathbb{E}_{\mathrm{p}_{tr}(\boldsymbol{y} | \boldsymbol{x})}\big[-\ln \mathrm{p}(\boldsymbol{y} | \boldsymbol{\mu}, \boldsymbol{\Lambda})\big]\big] \tag{23}$$

This will yield an upper bound on the cross entropy between the predictive posterior and the true distribution. However, if we were to consider the *forward* KL-divergence between Normal-Wishart distributions, we would not obtain such an expression and not correctly estimate data uncertainty. Interestingly, the reverse KL-divergence training objective has the same form as an ELBO - the predictive term and a reverse KL-divergence to the prior.

Having established an important property of the RKL objective, we not derive it's closed form expression. Note, that in these derivations, we make extended use of properties for taking expectations

of traces and log-determinants matrices with respect to the Wishart distribution detailed in (Gupta & Srivastava, 2010).For the first term in 22, we use the following property of the multivariate normal:

$$x \sim \mathcal{N}(\boldsymbol{\mu}, \boldsymbol{\Sigma}) \Rightarrow \mathbb{E}[\boldsymbol{x}^T \boldsymbol{A} \boldsymbol{x}] = \mathrm{Tr}(\boldsymbol{A}\boldsymbol{\Sigma}) + \boldsymbol{m}^T \boldsymbol{A} \boldsymbol{m}; \tag{24}$$

which allows us to get:

$$
\begin{aligned}
&\mathbb{E}_{\mathrm{p}(\boldsymbol{\mu},\boldsymbol{\Lambda}|\boldsymbol{x};\boldsymbol{\theta})}[-\ln \mathrm{p}(\boldsymbol{y}|\boldsymbol{\mu},\boldsymbol{\Lambda})] = \\
&= \frac{1}{2}\mathbb{E}_{\mathrm{p}(\boldsymbol{\mu},\boldsymbol{\Lambda}|\boldsymbol{x};\boldsymbol{\theta})}[(\boldsymbol{y}-\boldsymbol{\mu})^T\boldsymbol{\Lambda}(\boldsymbol{y}-\boldsymbol{\mu}) + K\ln(2\pi) - \ln|\boldsymbol{\Lambda}|] \\
&= \frac{1}{2}\mathbb{E}_{\mathcal{N}(\boldsymbol{\mu}|\boldsymbol{m},\kappa\boldsymbol{\Lambda})\mathcal{W}(\boldsymbol{\Lambda}|\boldsymbol{L},\nu)}[(\boldsymbol{y}-\boldsymbol{\mu})^T\boldsymbol{\Lambda}(\boldsymbol{y}-\boldsymbol{\mu}) + K\ln(2\pi) - \ln|\boldsymbol{\Lambda}|] \\
&\propto \frac{1}{2}\mathbb{E}_{\mathrm{p}(\boldsymbol{\Lambda}|L;\nu)}[(\boldsymbol{y}-\boldsymbol{m})^T\boldsymbol{\Lambda}(\boldsymbol{y}-\boldsymbol{m}) + K\kappa^{-1} - \ln|\boldsymbol{\Lambda}|] \\
&= \frac{\nu}{2}(\boldsymbol{y}-\boldsymbol{m})^T\boldsymbol{L}(\boldsymbol{y}-\boldsymbol{m}) + \frac{K}{2\kappa} - \frac{1}{2}\ln|\boldsymbol{L}| - \frac{1}{2}\psi_K(\frac{\nu}{2}) + \frac{K}{2}\ln\pi.
\end{aligned}
\tag{25}
$$

The second term in (22) may expressed as follows via the chain-rule of relative entropy(Cover & Thomas, 2006):

$$
\begin{aligned}
&\mathrm{KL}[\mathrm{p}(\boldsymbol{\mu},\boldsymbol{\Lambda}|\boldsymbol{\Omega})\|\mathrm{p}(\boldsymbol{\mu},\boldsymbol{\Lambda}|\boldsymbol{\Omega}_0)] = \\
&= \mathbb{E}_{\mathrm{p}(\boldsymbol{\Lambda}|\boldsymbol{\Omega})}[\mathrm{KL}[\mathrm{p}(\boldsymbol{\mu}|\boldsymbol{\Lambda},\boldsymbol{\Omega})\|\mathrm{p}(\boldsymbol{\mu}|\boldsymbol{\Lambda},\boldsymbol{\Omega}_0)]] + \mathrm{KL}[\mathrm{p}(\boldsymbol{\Lambda}|\boldsymbol{\Omega})\|\mathrm{p}(\boldsymbol{\Lambda}|\boldsymbol{\Omega}_0)];
\end{aligned}
\tag{26}
$$

The first term in (26) can be computed as:

$$
\begin{aligned}
&\mathbb{E}_{\mathrm{p}(\boldsymbol{\Lambda}|\boldsymbol{\Omega})}[\mathrm{KL}[\mathrm{p}(\boldsymbol{\mu}|\boldsymbol{\Lambda},\boldsymbol{\Omega})\|\mathrm{p}(\boldsymbol{\mu}|\boldsymbol{\Lambda},\boldsymbol{\Omega}_0)]] = \\
&= \mathbb{E}_{\mathcal{W}(\boldsymbol{\Lambda}|\boldsymbol{L},\nu)}[\mathrm{KL}[\mathcal{N}(\boldsymbol{y}|\boldsymbol{m},\kappa\boldsymbol{\Lambda})\|\mathcal{N}(\boldsymbol{y}|\boldsymbol{m}_0,\kappa_0\boldsymbol{\Lambda})]] \\
&= \frac{\kappa_0}{2}(\boldsymbol{m}-\boldsymbol{m}_0)^T\nu\boldsymbol{L}(\boldsymbol{m}-\boldsymbol{m}_0) + \frac{K}{2}(\frac{\kappa_0}{\kappa} - \ln\frac{\kappa_0}{\kappa} - 1);
\end{aligned}
\tag{27}
$$

while the second term is:

$$
\begin{aligned}
&\mathrm{KL}[\mathrm{p}(\boldsymbol{\Lambda}|\boldsymbol{\Omega})\|\mathrm{p}(\boldsymbol{\Lambda}|\boldsymbol{\Omega}_0)] = \mathrm{KL}[\mathcal{W}(\boldsymbol{\Lambda}|\boldsymbol{L},\nu)\|\mathcal{W}(\boldsymbol{\Lambda}|\boldsymbol{L}_0,\nu_0)] = \\
&= \frac{\nu}{2}\big(\mathrm{tr}(\boldsymbol{L}_0^{-1}\boldsymbol{L}) - K\big) - \frac{\nu_0}{2}\ln|\boldsymbol{L_0}^{-1}\boldsymbol{L}| + \ln\frac{\Gamma_K(\frac{\nu_0}{2})}{\Gamma_K(\frac{\nu}{2})} + \frac{\nu-\nu_0}{2}\psi_K(\frac{\nu}{2}).
\end{aligned}
\tag{28}
$$

### A.3 UNCERTAINTY MEASURES

Given a Normal-Wishart Prior Network which displays the desired set of behaviours detailed in 2, it is possible to compute closed-form expression for all measures of uncertainty previously discussed for Dirichlet Prior Networks (Malinin, 2019). The current section details the derivations of uncertainty measures introduced in section 2 for the Normal-Wishart distribution. We make extensive use of (Gupta & Srivastava, 2010) for taking expectation of log-determinants and traces of matrices.

#### A.3.1 DIFFERENTIAL ENTROPY OF PREDICTIVE POSTERIOR

As discussed in section 2, the predictive posterior of a Prior Network which parameterizes a Normal-Wishart distribution is a multivariate T distribution:

$$\mathbb{E}_{\mathrm{p}(\boldsymbol{\mu},\boldsymbol{\Lambda}|\boldsymbol{x},\boldsymbol{\theta})}[\mathrm{p}(\boldsymbol{y}|\boldsymbol{\mu},\boldsymbol{\Lambda})] = \mathcal{T}(\boldsymbol{y}|\boldsymbol{m}, \frac{\kappa+1}{\kappa(\nu-K+1)}\boldsymbol{L}^{-1}, \nu-K+1). \tag{29}$$

The differential entropy of the predictive posterior will be a measure of *total uncertainty*. The differential entropy of a standard multivariate student's T distribution with an identity scatter matrix $\boldsymbol{\Sigma} = \boldsymbol{I}$ is given by:

$$\mathcal{H}[\mathcal{T}(\boldsymbol{x}|\boldsymbol{\mu},\boldsymbol{I},\nu)] = -\ln\frac{\Gamma(\frac{\nu+K}{2})}{\Gamma(\frac{\nu}{2})(\nu\pi)^{\frac{K}{2}}} + (\frac{\nu+K}{2}) \cdot (\psi(\frac{\nu+K}{2}) - \psi(\frac{\nu}{2})); \tag{30}$$

which is a result obtained from (ARELLANO-VALLE et al., 2013). Using the property of differential entropy (Cover & Thomas, 2006), that if $\boldsymbol{x} \sim \mathrm{p}(\boldsymbol{x})$ and $\boldsymbol{y} = \boldsymbol{\mu} + \boldsymbol{A}\boldsymbol{x}$, then:

$$\mathcal{H}[\mathrm{p}(\boldsymbol{y})] = \mathcal{H}[\mathrm{p}(\boldsymbol{x})] + \ln|\boldsymbol{A}|. \tag{31}$$

We can show that the differential entropy of a standard general multivariate student's T distribution is given by:

$$\mathcal{H}[\mathcal{T}(\boldsymbol{x}|\boldsymbol{\mu}, \boldsymbol{\Sigma}, \nu)] = \frac{1}{2}\ln|\boldsymbol{\Sigma}| - \ln\frac{\Gamma(\frac{\nu+K}{2})}{\Gamma(\frac{\nu}{2})(\nu\pi)^{\frac{K}{2}}} + (\frac{\nu+K}{2})\cdot\big(\psi(\frac{\nu+K}{2}) - \psi(\frac{\nu}{2})\big). \tag{32}$$

Using this expression, we can show that the differential entropy of the predictive posterior of a Normal-Wishart Prior Network is given by:

$$\begin{aligned}
\mathcal{H}\big[\mathbb{E}_{\mathrm{p}(\boldsymbol{\mu},\boldsymbol{\Lambda}|\boldsymbol{x},\boldsymbol{\theta})}[\mathrm{p}(\boldsymbol{y}|\boldsymbol{\mu},\boldsymbol{\Lambda})]\big] &= \mathcal{H}\big[\mathcal{T}(\boldsymbol{y}|\boldsymbol{m}, \frac{\kappa+1}{\kappa(\nu-K+1)}\boldsymbol{L}^{-1}, \nu-K+1)\big] \\
&= \frac{\nu+1}{2}\big(\psi(\frac{\nu+1}{2}) - \psi(\frac{\nu-K+1}{2})\big) - \ln\frac{\Gamma(\frac{\nu+1}{2})}{\Gamma(\frac{\nu-K+1}{2})\big((\nu-K+1)\pi\big)^{\frac{K}{2}}} \\
&\quad - \frac{1}{2}\ln|\boldsymbol{L}| + \frac{K}{2}\ln\frac{\kappa+1}{\kappa(\nu-K+1)}.
\end{aligned} \tag{33}$$

### A.3.2 MUTUAL INFORMATION

The mutual information between the target $\boldsymbol{y}$ and the parameters of the output distribution $\{\boldsymbol{\mu}, \boldsymbol{\Lambda}\}$ is measures of *knowledge uncertainty*, and it the difference between the (differential) entropy of the predictive posterior and the expected differential entropy of each normal distribution sampled from the Normal Wishart:

$$\underbrace{\mathcal{I}[\boldsymbol{y}, \{\boldsymbol{\mu}, \boldsymbol{\Lambda}\}]}_{\text{Knowledge Uncertainty}} = \underbrace{\mathcal{H}\big[\mathbb{E}_{\mathrm{p}(\boldsymbol{\mu},\boldsymbol{\Lambda}|\boldsymbol{x},\boldsymbol{\theta})}[\mathrm{p}(\boldsymbol{y}|\boldsymbol{\mu},\boldsymbol{\Lambda})]\big]}_{\text{Total Uncertainty}} - \underbrace{\mathbb{E}_{\mathrm{p}(\boldsymbol{\mu},\boldsymbol{\Lambda}|\boldsymbol{x},\boldsymbol{\theta})}\big[\mathcal{H}[\mathrm{p}(\boldsymbol{y}|\boldsymbol{\mu},\boldsymbol{\Lambda})]\big]}_{\text{Expected Data Uncertainty}} \tag{34}$$

The first term, the differential entropy of the predictive posterior was derived above in (33). We we derive the expected differential entropy as follows:

$$\begin{aligned}
\mathbb{E}_{\mathcal{N}\mathcal{W}(\boldsymbol{\mu},\boldsymbol{\Lambda}|\Omega)}\big[\mathcal{H}[\mathcal{N}(\boldsymbol{y}|\boldsymbol{\mu},\boldsymbol{\Lambda})]\big] &= \frac{1}{2}\mathbb{E}_{\mathcal{N}\mathcal{W}(\boldsymbol{\mu},\boldsymbol{\Lambda}|\Omega)}\big[K\ln(2\pi e) - \ln|\boldsymbol{\Lambda}|\big] = \\
&= \frac{1}{2}\big[K\ln(\pi e) - \ln|\boldsymbol{L}| - \psi_K(\frac{\nu}{2}))\big].
\end{aligned} \tag{35}$$

Thus, the final expression for mutual information is:

$$\begin{aligned}
\mathcal{I}[\boldsymbol{y}, \{\boldsymbol{\mu}, \boldsymbol{\Lambda}\}] &= \frac{\nu+1}{2}\big(\psi(\frac{\nu+1}{2}) - \psi(\frac{\nu-K+1}{2})\big) - \ln\frac{\Gamma(\frac{\nu+1}{2})}{\Gamma(\frac{\nu-K+1}{2})\big((\nu-K+1)\pi\big)^{\frac{K}{2}}} \\
&\quad + \frac{K}{2}\ln\frac{\kappa+1}{\kappa(\nu-K+1)} - \frac{1}{2}\big[K\ln(\pi e) - \psi_K(\frac{\nu}{2}))\big].
\end{aligned} \tag{36}$$

Note that this expression is no longer a function of $\boldsymbol{L}$, which was important in representation *data uncertainty*.

### A.3.3 EXPECTED PAIRWISE KL-DIVERGENCE

An alternative measures of *knowledge uncertainty* which can be considered is the expected pairwise kl-divergence (EPKL), which upper bounds mutual information (Malinin, 2019). In this section we derive it's closed form expression for the Normal-Wishart distribution.

$$\begin{aligned}
\mathcal{K}[\boldsymbol{y}, \{\boldsymbol{\mu}, \boldsymbol{\Lambda}\}] &= \mathbb{E}_{\mathrm{p}(\boldsymbol{\mu}_0,\boldsymbol{\Lambda}_0)}\mathbb{E}_{p(\boldsymbol{\mu}_1,\boldsymbol{\Lambda}_1)}\big[\mathrm{KL}[\mathcal{N}(\boldsymbol{y}|\boldsymbol{\mu}_1,\boldsymbol{\Lambda}_1)\|\mathcal{N}(\boldsymbol{y}|\boldsymbol{\mu}_0,\boldsymbol{\Lambda}_0)]\big] \\
&= \frac{1}{2}\mathbb{E}_{\mathrm{p}(\boldsymbol{\mu}_0,\boldsymbol{\Lambda}_0)}\mathbb{E}_{p(\boldsymbol{\mu}_1,\boldsymbol{\Lambda}_1)}\big[(\boldsymbol{\mu}_1-\boldsymbol{\mu}_0)^T\boldsymbol{\Lambda}_0(\boldsymbol{\mu}_1-\boldsymbol{\mu}_0) + \ln\frac{|\boldsymbol{\Lambda}_1|}{|\boldsymbol{\Lambda}_0|} + \mathrm{Tr}(\boldsymbol{\Lambda}_0\boldsymbol{\Lambda}_1^{-1}) - K\big].
\end{aligned} \tag{37}$$

Here $\mathrm{p}(\boldsymbol{\mu}_0, \boldsymbol{\Lambda}_0) = \mathrm{p}(\boldsymbol{\mu}_1, \boldsymbol{\Lambda}_1) = \mathrm{p}(\boldsymbol{\mu}, \boldsymbol{\Lambda}|\boldsymbol{x}; \boldsymbol{\theta})$ . In (37) the first term is:

$$
\begin{aligned}
\mathbb{E}_{\mathrm{p}(\boldsymbol{\mu}_0, \boldsymbol{\Lambda}_0)}\mathbb{E}_{p(\boldsymbol{\mu}_1, \boldsymbol{\Lambda}_1)}&\big[(\boldsymbol{\mu}_1 - \boldsymbol{\mu}_0)^T \boldsymbol{\Lambda}_0 (\boldsymbol{\mu}_1 - \boldsymbol{\mu}_0)\big] = \\
&= \mathbb{E}_{\mathrm{p}(\boldsymbol{\mu}_0, \boldsymbol{\Lambda}_0)}\mathbb{E}_{p(\boldsymbol{\Lambda}_1)}\big[(\boldsymbol{m} - \boldsymbol{\mu}_0)^T \boldsymbol{\Lambda}_0 (\boldsymbol{m} - \boldsymbol{\mu}_0) + \mathrm{Tr}(\boldsymbol{\Lambda}_0 \tfrac{1}{\kappa}\boldsymbol{\Lambda}_1^{-1})\big] \\
&= \mathbb{E}_{\mathrm{p}(\boldsymbol{\mu}_0, \boldsymbol{\Lambda}_0)}\Big[(\boldsymbol{m} - \boldsymbol{\mu}_0)^T \boldsymbol{\Lambda}_0 (\boldsymbol{m} - \boldsymbol{\mu}_0) + \frac{1}{\kappa(\nu - K - 1)} \mathrm{Tr}(\boldsymbol{\Lambda}_0 L^{-1})\Big] \\
&= \frac{K}{\kappa} + \frac{\nu K}{\kappa(\nu - K - 1)};
\end{aligned}
\tag{38}
$$

The second term in (37) is zero, and the third term is:

$$
\mathbb{E}_{\mathrm{p}(\boldsymbol{\mu}_0, \boldsymbol{\Lambda}_0)}\mathbb{E}_{p(\boldsymbol{\mu}_1, \boldsymbol{\Lambda}_1)}\big[\mathrm{Tr}(\boldsymbol{\Lambda}_0 \boldsymbol{\Lambda}_1^{-1})\big] = \mathbb{E}_{\mathrm{p}(\boldsymbol{\mu}_0, \boldsymbol{\Lambda}_0)}\Big[\mathrm{Tr}(\boldsymbol{\Lambda}_0 \frac{1}{\nu - K - 1}L^{-1})\Big] = \frac{\nu K}{(\nu - K - 1)}; \tag{39}
$$

which in sum gives us:

$$
\mathcal{K}[\boldsymbol{y}, \{\boldsymbol{\mu}, \boldsymbol{\Lambda}\}] = \frac{1}{2}\frac{\nu K(\kappa^{-1} + 1)}{(\nu - K - 1)} - \frac{K}{2} + \frac{K}{2\kappa}. \tag{40}
$$

Note that this is also not a function of $L$, just like mutual information. Rather, it is only a function of the pseudo-counts $\kappa$ and $\nu$.

### A.3.4 LAW OF TOTAL VARIATION

Finally, in order to be able to compare with ensembles, we can also derive variance-based measures of *total*, *data* and *knowledge uncertainty* via the Law of total variance, as follows:

$$
\underbrace{\mathbb{V}_{\mathrm{p}(\boldsymbol{\mu}, \boldsymbol{\Lambda}|\boldsymbol{x}, \boldsymbol{\theta})}[\boldsymbol{\mu}]}_{\text{Knowledge Uncertainty}} = \underbrace{\mathbb{V}_{\mathrm{p}(\boldsymbol{y}|\boldsymbol{x}, \boldsymbol{\theta})}[\boldsymbol{y}]}_{\text{Total Uncertainty}} - \underbrace{\mathbb{E}_{\mathrm{p}(\boldsymbol{\mu}, \boldsymbol{\Lambda}|\boldsymbol{x}, \boldsymbol{\theta})}[\boldsymbol{\Lambda}^{-1}]}_{\text{Expected Data Uncertainty}} \tag{41}
$$

This has a similar decomposition as mutual information. In this section we derive its closed form expression. We can compute the expected variance by using the probabilistic change of variables:

$$
\begin{aligned}
\mathbb{E}_{\mathrm{p}(\boldsymbol{\mu}, \boldsymbol{\Lambda}|\boldsymbol{x}, \boldsymbol{\theta})}[\boldsymbol{\Lambda}^{-1}] &= \mathbb{E}_{\mathcal{NW}(\boldsymbol{\mu}, \boldsymbol{\Lambda})}[\boldsymbol{\Lambda}^{-1}] = \mathbb{E}_{\mathcal{W}(\boldsymbol{\Lambda})}[\boldsymbol{\Lambda}^{-1}] = \mathbb{E}_{\mathcal{W}^{-1}(\boldsymbol{\Lambda}^{-1})}[\boldsymbol{\Lambda}^{-1}] \\
&= \frac{1}{\nu - K - 1}L^{-1};
\end{aligned}
\tag{42}
$$

and the variance of expected as:

$$
\begin{aligned}
\mathbb{V}_{\mathrm{p}(\boldsymbol{\mu}, \boldsymbol{\Lambda}|\boldsymbol{x}, \boldsymbol{\theta})}[\boldsymbol{\mu}] &= \mathbb{E}_{\mathcal{NW}(\boldsymbol{\mu}, \boldsymbol{\Lambda})}\big[(\boldsymbol{\mu} - \boldsymbol{m})(\boldsymbol{\mu} - \boldsymbol{m})^T\big] = \frac{1}{\kappa}\mathbb{E}_{\mathcal{W}(\boldsymbol{\Lambda}|L, \nu)}[\boldsymbol{\Lambda}^{-1}] \\
&= \frac{1}{\kappa(\nu - K - 1)}L^{-1}.
\end{aligned}
\tag{43}
$$

Thus, the total variance is then expressed as:

$$
\mathbb{V}_{\mathrm{p}(\boldsymbol{y}|\boldsymbol{x}, \boldsymbol{\theta})}[\boldsymbol{y}] = \frac{1 + \kappa}{\kappa(\nu - K - 1)}L^{-1}. \tag{44}
$$

Note that this yields a measure which only considers first and second moments. In addition, in order to obtain a scalar estimate of uncertainty, it is necessary to consider the log-determinant of each measure.

## B EXPERIMENT ON SYNTHETIC DATA

The training data consists of 2048 inputs $x$ uniformly sampled from $[-10, 10]$ with targets $y \sim \mathcal{N}(\sin x + \frac{x}{10})$. We use a relu network with 2 hidden layers containing 30 units each to predict the parameters of either Gaussian or Normal-Wishart distribution on this data. In all cases, we use Adam optimizer with learning rate $10^{-2}$ and weight decay $10^{-4}$ for 800 epochs with batch size 128. Gaussian models in an ensemble are trained via negative log-likelihood starting from different random initialization. To train a Regression Prior Network with reverse-KL divergence 512 points were uniformly sampled from $[-25, -20] \cup [20, 25]$ as training-ood data. We use objective (8) with coefficient $\gamma = 0.5$. The prior belief is $\kappa_0 = 10^{-2}$ and in-domain $\hat{\beta}$ is $10^2$. For EnD$^2$ training we set $T = 1$ and add gaussian noise to inputs with standard deviation 3.

## C   UCI EXPERIMENTS

The current appendix provides additional details of experiments on the UCI regression datasets. Note that we leave out the Yacht Hydrodynamics datasets, as it is the smallest with the fewest features. The remaining datasets are described in the table below:

Table 5: Description of UCI datasets.

| Dataset | size | number of features |
|---|---|---|
| Boston housing | 506 | 13 |
| Concrete Compressive Strength | 1030 | 8 |
| Energy efficiency | 768 | 9 |
| Combined Cycle Power | 9568 | 4 |
| Red Wine Quality | 1599 | 11 |
| YearPredictionMSD | 515345 | 90 |

### C.1   TRAINING

Following (Lakshminarayanan et al., 2017) in all experiments except MSD we use 1 layer relu neural network with 50 hidden units, for MSD we use 100 hidden units. We optimize weights with Adam for 100 epochs with batch size 32. All hyper-parameters, including learning rate, weight decay, RKL prior belief in train data $\kappa_0$, RKL OOD coefficient $\gamma$, EnD$^2$ initial temperature $T$ and noise level $\varepsilon$ are set based on grid search, where we use an equal computational budget in all models to ensure a fair comparison. Additionally, we use 10 folds cross-validation and report standard deviation based on it.

### C.2   CREATING OUT-OF-DOMAIN DATA

Here we detail how OOD training data for reverse KL-divergence trained Prior Networks is created and how evaluation OOD data is created.

As reverse KL-divergence training of Prior Networks requires out-of-domain training examples, we use a factor analysis model to generate samples for this in the same way as was done in (Malinin, 2019). Specifically, we train a linear generative model that approximates inputs with $\hat{x} \sim \mathcal{N}(\mu, WW^T + \Psi)$, where $[W, \mu, \Psi]$ are model parameters. In-domain training data is used to train this model. The out-of-domain training examples are then sampled from $\mathcal{N}(\mu, 3WW^T + 3\Psi)$.

To estimate the quality of out-of-domain detection, we additionally create evaluation out-of-domain data from external UCI datasets: "Relative location of CT slices on axial axis Data Set" for MSD and "Condition Based Maintenance of Naval Propulsion Plants Data Set" for other datasets. We drop all constant columns in them and leave first $K$ columns and first $N$ rows, where $K$ is a number of features and $N$ is a number of test examples in a corresponding dataset. For each comparison, the out-of-domain datasets are normalized by the per-column mean and variance obtained on in-domain training data, in order to make out-of-domain detection task more difficult.

### C.3   PREDICTION REJECTION RATIO

Prediction Rejection Ratio (PRR) (Malinin et al., 2020) measures what part of the best possible error-detection performance our algorithm covers. In Figure 4 prediction-rejection curves are shown. They are built by iteratively rejecting test examples according to different orders and computing MSE error on each step. Uncertainty curve corresponds to the order of decreasing uncertainty of some model. Oracle curve corresponds to the best possible order. Prediction Rejection Ration is a ratio of the area between Uncertainty and Random curves $AR_{uncertainty}$ (orange in Figure 4) and the area between Oracle and Random curves $AR_{oracle}$ (blue in Figure 4):

$$PRR = \frac{AR_{uncertainty}}{AR_{oracle}}.$$

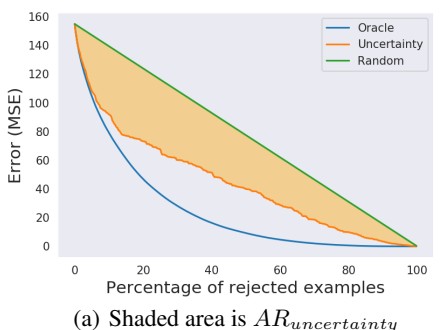
(a) Shaded area is $AR_{uncertainty}$

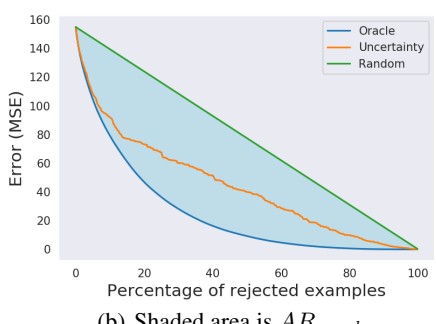
(b) Shaded area is $AR_{oracle}$

Figure 4: Prediction-rejection curves.

## C.4 RESULTS FOR ALL DATASETS

The current section provide a full set of predictive performance, error detection and OOD detection results on UCI datasets in tables 6-8, respectively. Results in table 6 show that all models achieve comparable performance and that EnD$^2$ tends to come close to the ensemble. Table 7 shows the error detection performance of all models in terms of prediction-rejection ratio (PRR). The results clearly show that measures of *total uncertainty* are useful in detecting errors, though it is more challenging on some datasets. At the same time, measures of *knowledge uncertainty* do significantly worse. Finally, table 8 shows the OOD detection performance in terms of % AUC-ROC. Here measures of *knowledge uncertainty* do far better. Notably, on the larger datasets, EnD$^2$ comes closer to the performance of the ensemble.

Table 6: Prediction performance metrics of models on six UCI datasets.

| Data | RMSE ($\downarrow$) | | | | NLL ($\downarrow$) | | | |
|------|--------|------|--------|------|--------|------|--------|------|
| | Single | ENSM | EnD$^2$ | NWPN | Single | ENSM | EnD$^2$ | NWPN |
| boston | $3.54 \pm 0.32$ | $\mathbf{3.52 \pm 0.32}$ | $3.64 \pm 0.33$ | $3.53 \pm 0.31$ | $2.58 \pm 0.08$ | $2.53 \pm 0.07$ | $2.5 \pm 0.05$ | $\mathbf{2.47 \pm 0.04}$ |
| energy | $1.83 \pm 0.07$ | $\mathbf{1.79 \pm 0.07}$ | $1.98 \pm 0.05$ | $1.83 \pm 0.07$ | $1.38 \pm 0.04$ | $\mathbf{1.32 \pm 0.04}$ | $1.72 \pm 0.04$ | $1.65 \pm 0.08$ |
| concrete | $5.66 \pm 0.19$ | $\mathbf{5.24 \pm 0.18}$ | $6.13 \pm 0.18$ | $5.77 \pm 0.24$ | $3.11 \pm 0.08$ | $\mathbf{3.00 \pm 0.04}$ | $3.13 \pm 0.03$ | $3.05 \pm 0.03$ |
| wine | $0.65 \pm 0.01$ | $\mathbf{0.63 \pm 0.01}$ | $\mathbf{0.63 \pm 0.01}$ | $\mathbf{0.63 \pm 0.01}$ | $1.24 \pm 0.16$ | $0.96 \pm 0.03$ | $\mathbf{0.91 \pm 0.02}$ | $0.93 \pm 0.02$ |
| power | $4.07 \pm 0.07$ | $\mathbf{4.00 \pm 0.07}$ | $4.06 \pm 0.07$ | $4.09 \pm 0.07$ | $2.82 \pm 0.02$ | $\mathbf{2.79 \pm 0.02}$ | $2.79 \pm 0.01$ | $2.81 \pm 0.01$ |
| MSD | $9.08 \pm 0.00$ | $\mathbf{8.92 \pm 0.00}$ | $8.94 \pm 0.00$ | $9.07 \pm 0.00$ | $3.51 \pm 0.00$ | $\mathbf{3.39 \pm 0.00}$ | $\mathbf{3.39 \pm 0.00}$ | $3.41 \pm 0.00$ |

Table 7: PRR scores on all six UCI datasets.

| Data | Model | $\mathcal{H}[\mathbb{E}]$ | $\mathbb{V}[\boldsymbol{y}]$ | $\mathcal{I}$ | $\mathcal{K}$ | $\mathbb{V}[\boldsymbol{\mu}]$ |
|------|-------|---------------------------|------------------------------|---------------|---------------|--------------------------------|
| boston | ENSM | - | $0.60 \pm 0.07$ | - | $-0.15 \pm 0.07$ | $0.41 \pm 0.08$ |
| | EnD$^2$ | $\mathbf{0.61 \pm 0.07}$ | $\mathbf{0.61 \pm 0.07}$ | $-0.02 \pm 0.15$ | $-0.02 \pm 0.15$ | $0.59 \pm 0.06$ |
| | NWPN | $0.54 \pm 0.08$ | $0.54 \pm 0.08$ | $-0.05 \pm 0.09$ | $-0.05 \pm 0.09$ | $0.49 \pm 0.08$ |
| energy | ENSM | - | $\mathbf{0.90 \pm 0.01}$ | - | $-0.80 \pm 0.02$ | $0.34 \pm 0.09$ |
| | EnD$^2$ | $0.83 \pm 0.02$ | $0.83 \pm 0.02$ | $-0.77 \pm 0.03$ | $-0.77 \pm 0.03$ | $0.60 \pm 0.05$ |
| | NWPN | $0.85 \pm 0.02$ | $0.85 \pm 0.02$ | $0.32 \pm 0.11$ | $0.32 \pm 0.11$ | $0.84 \pm 0.02$ |
| concrete | ENSM | - | $0.48 \pm 0.05$ | - | $0.01 \pm 0.07$ | $0.27 \pm 0.06$ |
| | EnD$^2$ | $0.50 \pm 0.03$ | $0.49 \pm 0.03$ | $0.05 \pm 0.06$ | $0.05 \pm 0.06$ | $0.44 \pm 0.03$ |
| | NWPN | $\mathbf{0.54 \pm 0.03}$ | $\mathbf{0.54 \pm 0.03}$ | $0.35 \pm 0.04$ | $0.35 \pm 0.04$ | $0.51 \pm 0.03$ |
| wine | ENSM | - | $\mathbf{0.32 \pm 0.02}$ | - | $0.10 \pm 0.03$ | $0.25 \pm 0.04$ |
| | EnD$^2$ | $0.30 \pm 0.02$ | $0.30 \pm 0.02$ | $0.06 \pm 0.03$ | $0.06 \pm 0.03$ | $0.30 \pm 0.02$ |
| | NWPN | $0.30 \pm 0.03$ | $0.30 \pm 0.03$ | $-0.19 \pm 0.03$ | $-0.19 \pm 0.03$ | $0.26 \pm 0.04$ |
| power | ENSM | - | $\mathbf{0.23 \pm 0.01}$ | - | $-0.01 \pm 0.02$ | $0.05 \pm 0.02$ |
| | EnD$^2$ | $\mathbf{0.23 \pm 0.01}$ | $\mathbf{0.23 \pm 0.01}$ | $-0.02 \pm 0.03$ | $-0.02 \pm 0.03$ | $0.15 \pm 0.02$ |
| | NWPN | $0.20 \pm 0.02$ | $0.20 \pm 0.02$ | $-0.03 \pm 0.02$ | $-0.03 \pm 0.02$ | $0.16 \pm 0.01$ |
| msd | ENSM | - | $\mathbf{0.64 \pm 0.0}$ | - | $0.07 \pm 0.0$ | $0.39 \pm 0.0$ |
| | EnD$^2$ | $0.63 \pm 0.0$ | $0.63 \pm 0.0$ | $0.04 \pm 0.0$ | $0.04 \pm 0.0$ | $0.59 \pm 0.0$ |
| | NWPN | $\mathbf{0.64 \pm 0.0}$ | $\mathbf{0.64 \pm 0.0}$ | $-0.20 \pm 0.0$ | $-0.20 \pm 0.0$ | $0.61 \pm 0.0$ |

Table 8: OOD Detection (ROC-AUC) of models on UCI datasets.

| Data | Model | $\mathcal{H}[\mathbb{E}]$ | $\mathbb{V}[\boldsymbol{y}]$ | $\mathcal{I}$ | $\mathcal{K}$ | $\mathbb{V}[\boldsymbol{\mu}]$ |
|---|---|---|---|---|---|---|
| boston | ENSM | - | $0.75 \pm 0.02$ | - | $0.68 \pm 0.02$ | $\mathbf{0.86 \pm 0.02}$ |
|  | EnD$^2$ | $0.75 \pm 0.01$ | $0.75 \pm 0.01$ | $0.63 \pm 0.01$ | $0.63 \pm 0.01$ | $0.76 \pm 0.01$ |
|  | NWPN | $0.64 \pm 0.02$ | $0.65 \pm 0.02$ | $0.71 \pm 0.01$ | $0.71 \pm 0.01$ | $0.69 \pm 0.01$ |
| energy | ENSM | - | $0.76 \pm 0.01$ | - | $0.51 \pm 0.03$ | $\mathbf{0.77 \pm 0.03}$ |
|  | EnD$^2$ | $0.61 \pm 0.02$ | $0.61 \pm 0.02$ | $0.57 \pm 0.02$ | $0.57 \pm 0.02$ | $0.63 \pm 0.03$ |
|  | NWPN | $0.43 \pm 0.02$ | $0.43 \pm 0.02$ | $0.66 \pm 0.01$ | $0.66 \pm 0.01$ | $0.47 \pm 0.02$ |
| concrete | ENSM | - | $\mathbf{0.84 \pm 0.01}$ | - | $0.78 \pm 0.02$ | $0.82 \pm 0.01$ |
|  | EnD$^2$ | $0.48 \pm 0.02$ | $0.48 \pm 0.02$ | $0.45 \pm 0.01$ | $0.45 \pm 0.01$ | $0.47 \pm 0.01$ |
|  | NWPN | $\mathbf{0.84 \pm 0.01}$ | $\mathbf{0.84 \pm 0.01}$ | $0.8 \pm 0.01$ | $0.8 \pm 0.01$ | $0.83 \pm 0.01$ |
| wine | ENSM | - | $0.48 \pm 0.03$ | - | $0.58 \pm 0.01$ | $0.56 \pm 0.02$ |
|  | EnD$^2$ | $0.59 \pm 0.03$ | $0.60 \pm 0.03$ | $\mathbf{0.65 \pm 0.01}$ | $\mathbf{0.65 \pm 0.01}$ | $\mathbf{0.65 \pm 0.03}$ |
|  | NWPN | $0.42 \pm 0.02$ | $0.42 \pm 0.02$ | $0.64 \pm 0.01$ | $0.64 \pm 0.01$ | $0.53 \pm 0.02$ |
| power | ENSM | - | $0.35 \pm 0.02$ | - | $0.64 \pm 0.02$ | $0.62 \pm 0.02$ |
|  | EnD$^2$ | $0.37 \pm 0.03$ | $0.37 \pm 0.03$ | $\mathbf{0.66 \pm 0.01}$ | $\mathbf{0.66 \pm 0.01}$ | $0.50 \pm 0.02$ |
|  | NWPN | $0.24 \pm 0.01$ | $0.24 \pm 0.01$ | $0.56 \pm 0.01$ | $0.56 \pm 0.01$ | $0.31 \pm 0.02$ |
| msd | ENSM | - | $0.66 \pm 0.0$ | - | $0.55 \pm 0.0$ | $0.62 \pm 0.0$ |
|  | EnD$^2$ | $0.66 \pm 0.0$ | $0.66 \pm 0.0$ | $0.50 \pm 0.0$ | $0.50 \pm 0.0$ | $0.65 \pm 0.0$ |
|  | NWPN | $0.68 \pm 0.0$ | $0.68 \pm 0.0$ | $0.73 \pm 0.0$ | $0.73 \pm 0.0$ | $\mathbf{0.75 \pm 0.0}$ |

## D  DEPTH ESTIMATION EXPERIMENTS

### D.1  METRICS DEFINITION

In this section we give a description for metrics in Table 3, which are commonly used in prior work on monocular depth estimation (Eigen et al., 2014; Fu et al., 2018; Alhashim & Wonka, 2018).

Let $\boldsymbol{y}$ be target depth map for a particular image, while $\hat{\boldsymbol{y}}$ is a predicted depth map, and let $\hat{\boldsymbol{y}}_i$ be a prediction for $i$-th pixel, while $I$ is the set of all pixels. Then, the metrics for an individual image are defined as follows:

$$
\begin{aligned}
&\text{Thresholds } (\delta_1, \delta_2, \delta_3): \% \text{ of } \hat{\boldsymbol{y}}_i \text{ s.t.} \max\left(\frac{\hat{\boldsymbol{y}}_i}{\boldsymbol{y}_i}, \frac{\boldsymbol{y}_i}{\hat{\boldsymbol{y}}_i}\right) < \delta^k, \ \delta = 1.25, \ k = 1, 2, 3; \\
&\text{Absolute Relative Difference (rel): } \frac{1}{|I|} \sum_{i \in I} \frac{|\hat{\boldsymbol{y}}_i - \boldsymbol{y}_i|}{\boldsymbol{y}_i}; \\
&\text{RMSE in log space: } (\log_{10}) \sqrt{\frac{1}{|I|} \sum_{i \in I} (\log_{10} \hat{\boldsymbol{y}}_i - \log_{10} \boldsymbol{y}_i)^2}.
\end{aligned}
\tag{45}
$$

Metrics for the whole dataset are obtained by averaging individual metrics. They allow to assess different properties of the model: $\delta_k$ demonstrate confidence intervals, rel shows the ratio between prediction error and target, and $\log_{10}$ measures the error in log-space which is less sensitive to outliers.

### D.2  CALIBRATION

KDE estimates for histograms of NLLs are provided in the figure below. We can see that both EnD$^2$, DER, and NWPN models yield far more consistent NLL than their Gaussian counterparts, while the latter has a long tail of outliers. This is probably connected with the fact that the Student $\mathcal{T}$ distribution has heavier tails than Gaussian, which allocates greater probability mass farther from the mean, allowing the model to be less confident about its predictions, especially outliers. We also see that DER likelihoods are shifted slightly to the right, which highlights the bias introduced by its evidence regularizer.

### D.3  ADDITIONAL DEPTH ESTIMATION EXPERIMENTS

In RPN training with RKL objective, we observed the optimization trajectory to be very unstable and initialization-sensitive. To combat this, we linearly increase $\gamma$ from 0 to some predefined value during

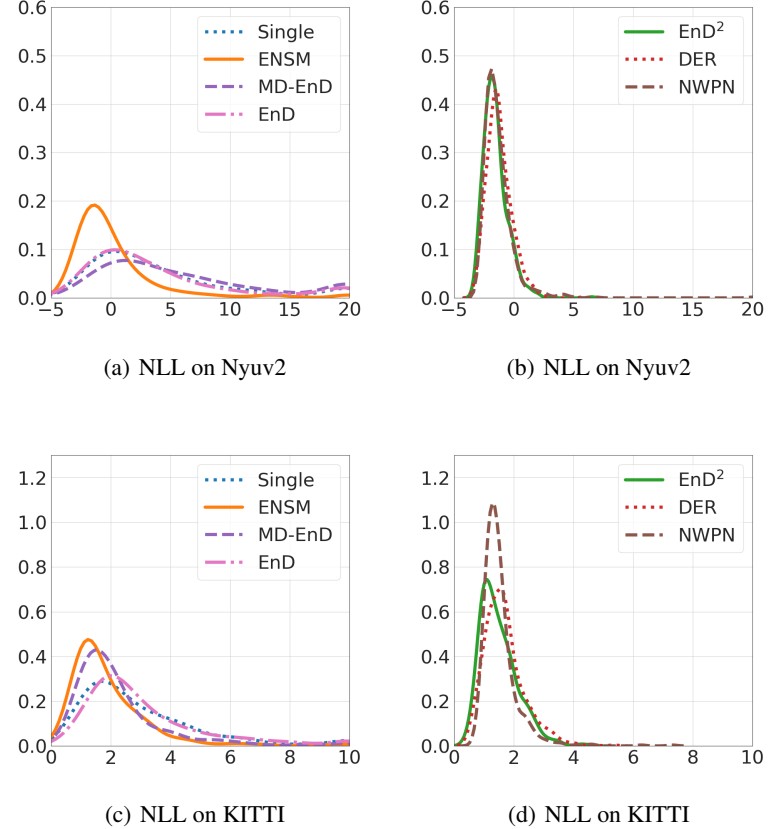

Figure 5: NLL histograms on Nyu and KITTI datasets.

the first five epochs, which allows our models to concentrate initially on predictive performance, and then gradually capture the properties of "in-domain" samples.

Additionally, we performed an ablation study across different coefficients $\gamma$, with the results provided in tables 9 and 10. On NYU dataset, we see that models with lower $\gamma$ improve performance at the cost of decreased quality of OOD detection. This may be an indication that the task of accurate prediction may not align well with the model's ability to detect unfamiliar samples. Based on this, we decided to use the coefficient of $\gamma = 0.05$ as achieving the best trade-off, and then fine-tuned the respective model for 10 additional epochs until convergence.

Table 9: RKL ablation on NYU with training OOD KITTI.

| Method | Predictive Performance | | | | | | | OOD Detection |
| | $\delta_1(\uparrow)$ | $\delta_2(\uparrow)$ | $\delta_3(\uparrow)$ | rel$(\downarrow)$ | rmse$(\downarrow)$ | $\log_{10}(\downarrow)$ | NLL$(\downarrow)$ | ROC-AUC$(\uparrow)(\mathcal{H}[\mathbb{E}])$ |
|---|---|---|---|---|---|---|---|---|
| Single | **0.852** | **0.971** | **0.993** | **0.122** | 0.456 | 0.053 | 5.74 | 0.69 |
| NWPN , $\gamma = 0$ | 0.852 | 0.971 | 0.993 | 0.122 | **0.450** | **0.052** | -1.46 | 0.691 |
| NWPN , $\gamma = 0.005$ | 0.819 | 0.957 | 0.987 | 0.148 | 0.496 | 0.06 | **-1.69** | 0.803 |
| NWPN , $\gamma = 0.01$ | 0.805 | 0.952 | 0.986 | 0.148 | 0.515 | 0.061 | -1.57 | 0.81 |
| NWPN , $\gamma = 0.025$ | 0.665 | 0.905 | 0.975 | 0.197 | 0.666 | 0.085 | -0.84 | 0.866 |
| NWPN , $\gamma = 0.05$ | 0.556 | 0.849 | 0.956 | 0.240 | 0.821 | 0.106 | 0.05 | **0.876** |

## D.4 ADDITIONAL OOD DETECTION EXPERIMENTS

We also provide additional OOD detection results, where models trained on NYU detect KITTI OOD data, and vice versa. Note, we do not evaluate NWPN in this scenario, as there two datasets represent the training data. The results generally follow the same trends as those outline in the main paper.

Table 10: RKL ablation on KITTI with training OOD NYU.

| Method | $\delta_1(\uparrow)$ | $\delta_2(\uparrow)$ | $\delta_3(\uparrow)$ | rel($\downarrow$) | rmse($\downarrow$) | $\log_{10}(\downarrow)$ | NLL($\downarrow$) | ROC-AUC($\uparrow$)($\mathcal{I}$) |
|---|---|---|---|---|---|---|---|---|
| | | | | | | | | OOD Detection |
| Single | 0.924 | 0.987 | 0.997 | 0.078 | 3.537 | 0.035 | 2.98 | 0.845 |
| NWPN , $\gamma = 0$ | 0.924 | 0.988 | 0.997 | 0.078 | 3.528 | 0.035 | 1.59 | 0.994 |
| NWPN , $\gamma = 0.005$ | 0.924 | 0.987 | 0.997 | 0.079 | 3.392 | 0.035 | 1.55 | 1.0 |
| NWPN , $\gamma = 0.01$ | 0.919 | 0.986 | 0.997 | 0.080 | 3.435 | 0.035 | 1.52 | 1.0 |
| NWPN , $\gamma = 0.025$ | 0.922 | 0.986 | 0.997 | 0.079 | 3.513 | 0.036 | 1.63 | 1.0 |
| NWPN , $\gamma = 0.05$ | 0.920 | 0.986 | 0.997 | 0.077 | 3.526 | 0.035 | 1.52 | 1.0 |

Table 11: OOD Detection (ROC-AUC) of NYU models on KITTI and vice-versa.

| Method | NYUv2 vs KITTI OOD Detect | | | | | KITTI vs NYUv2 OOD Detect | | | | |
|---|---|---|---|---|---|---|---|---|---|---|
| | $\mathcal{H}[\mathbb{E}]$ | $\mathbb{V}[\boldsymbol{y}]$ | $\mathcal{I}$ | $\mathcal{K}$ | $\mathbb{V}[\boldsymbol{\mu}]$ | $\mathcal{H}[\mathbb{E}]$ | $\mathbb{V}[\boldsymbol{y}]$ | $\mathcal{I}$ | $\mathcal{K}$ | $\mathbb{V}[\boldsymbol{\mu}]$ |
| Single | 0.892 | 0.892 | - | - | - | 0.013 | 0.013 | - | - | - |
| ENSM | - | 0.927 | - | 0.779 | 0.929 | - | 0.02 | - | **0.806** | 0.076 |
| EnD | 0.774 | 0.774 | - | - | - | 0.001 | 0.001 | - | - | - |
| EnD$^2$ | 0.896 | 0.905 | **0.946** | 0.938 | 0.935 | 0.004 | 0.004 | 0.750 | 0.724 | 0.009 |
| DER | 0.937 | 0.937 | 0.882 | 0.811 | 0.937 | 0.038 | 0.04 | 0.04 | 0.008 | 0.047 |
| MD-EnD | - | 0.707 | - | 0.223 | 0.403 | - | 0.002 | - | 0.733 | 0.026 |

Additionally, in the following two figures we provide examples of the in-domain and OOD data as it appear when it is pre-processed. From the figures we can clearly see that relative to NYU Depth, all images are either at comparable depth or a little farther. However, relative to KITTI, all OOD images are much closer to the camera, both naturally, and exacerbated via the crop and scale operation.

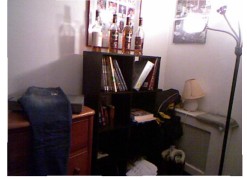 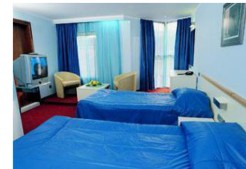 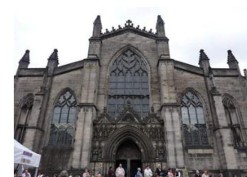 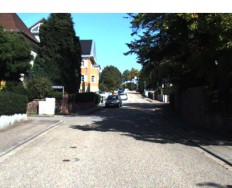

(a) NYU Depth V2 ID   (b) LSUN-BED OOD   (c) LSUN-Church OOD   (d) KITTI OOD

Figure 6: Examples of test inputs for Nyu model. Images are in order: NYU, LSUN-bed, LSUN-church, KITTI. OOD images are center-cropped and re-scaled to the in-domain data, preserving the aspect ratio.

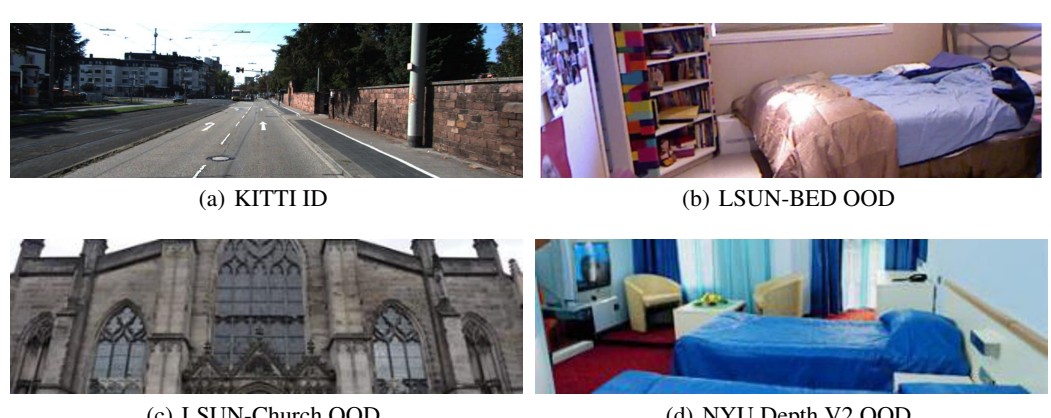

(a) KITTI ID

(b) LSUN-BED OOD

(c) LSUN-Church OOD

(d) NYU Depth V2 OOD

Figure 7: Examples of test inputs for KITTI model. Images are in order: KITTI, LSUN-bed, LSUN-church, NYU. OOD images are center-cropped and re-scaled to the in-domain data, preserving the aspect ratio.

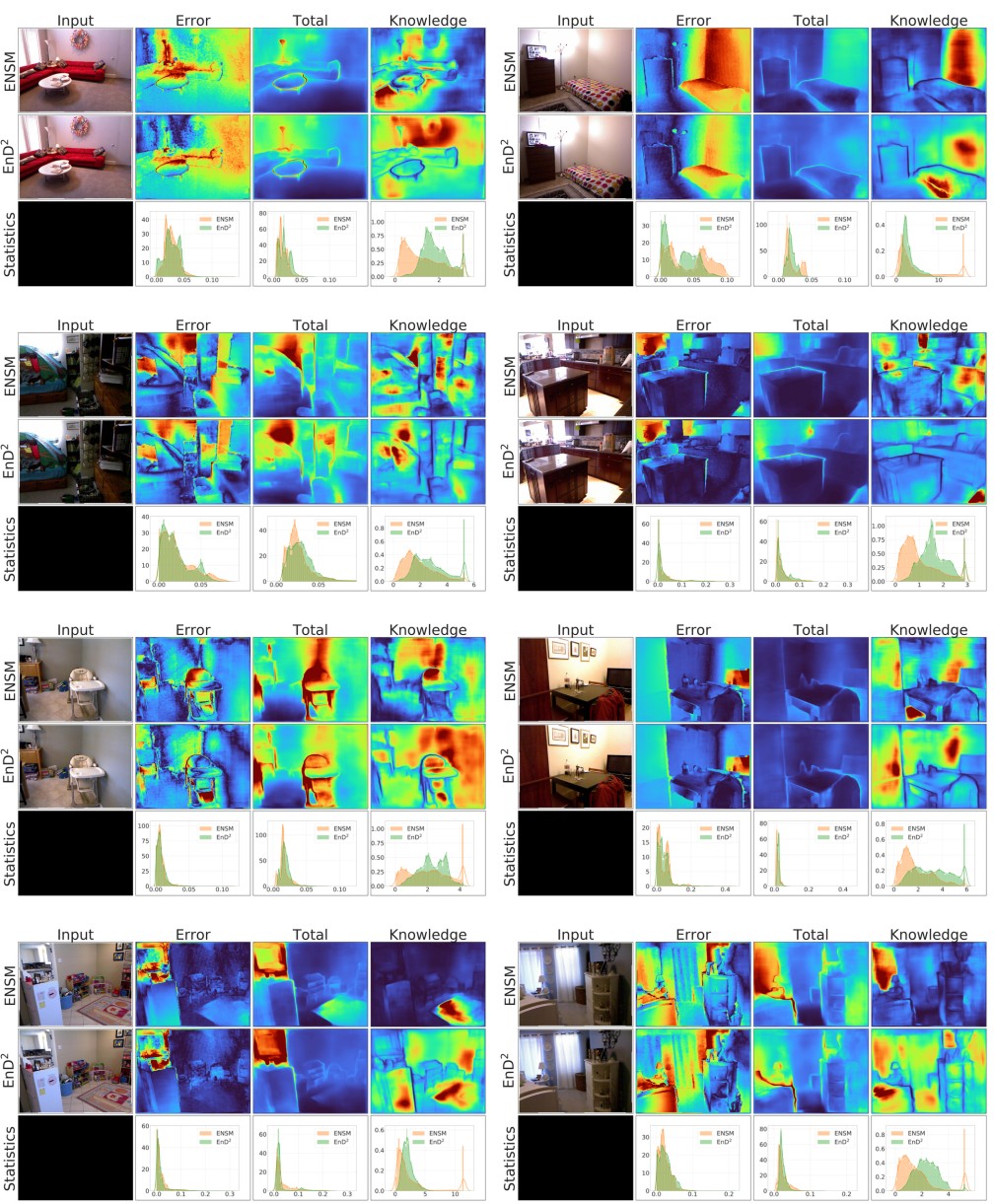

Figure 8: Uncurated comparison of ENSM vs EnD$^2$ behaviour on Nyuv2 dataset (best viewed in color).

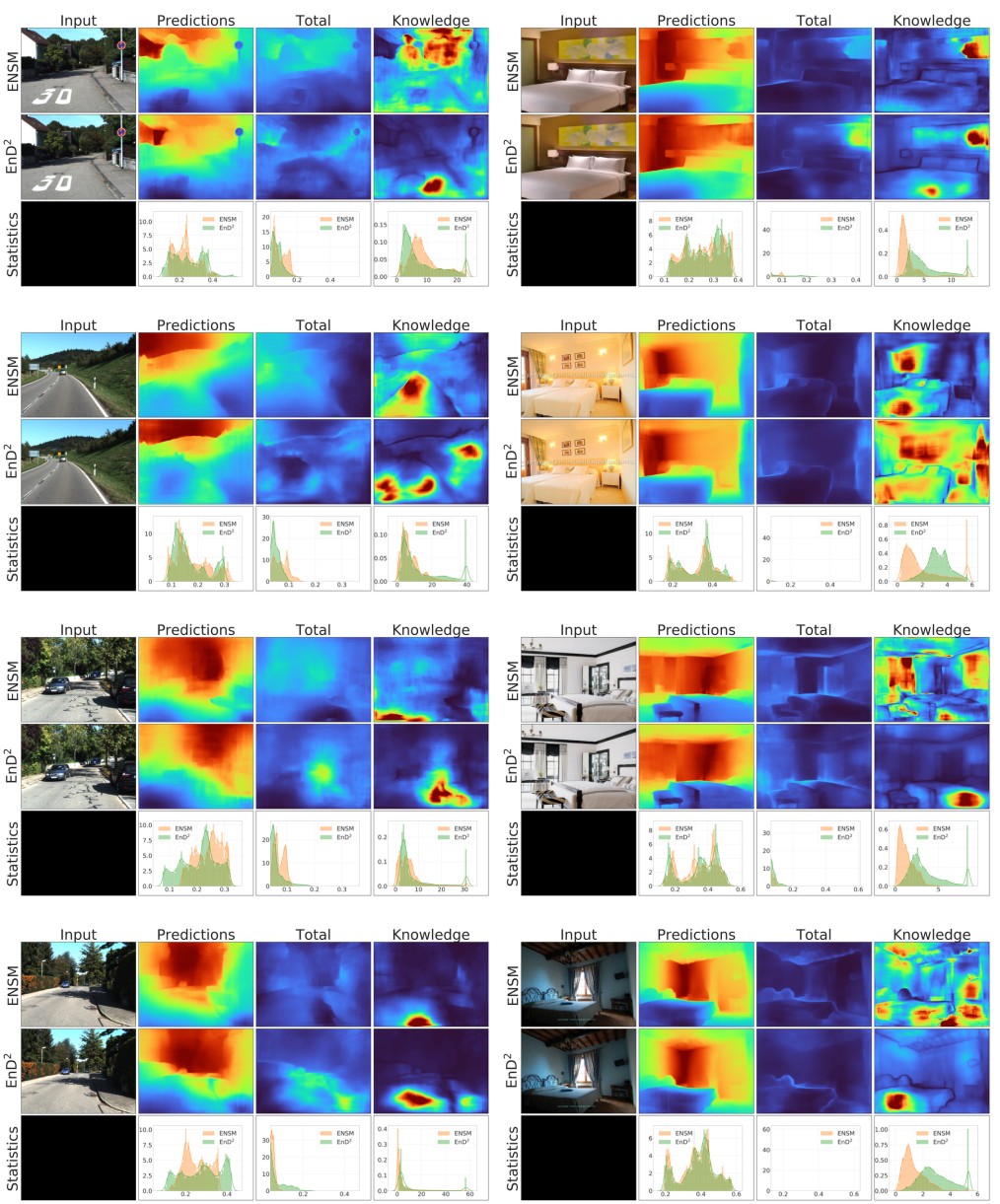

Figure 9: Uncurated comparison of ENSM vs EnD$^2$ models trained on Nyuv2 behaviour on Kitti and LSUN-bed datasets (best viewed in color).

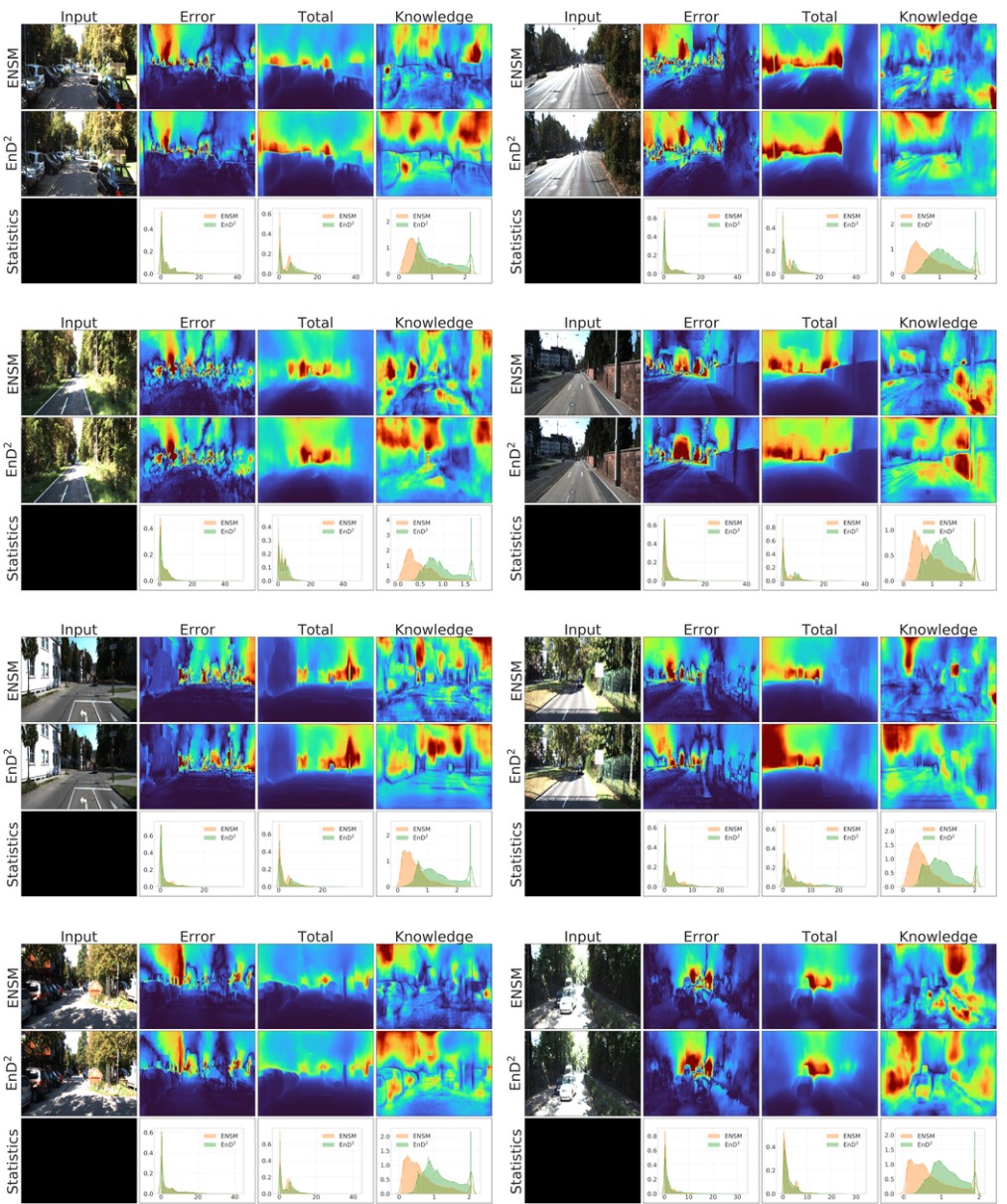

Figure 10: Uncurated comparison of ENSM vs EnD$^2$ behaviour on KITTI dataset (best viewed in color).

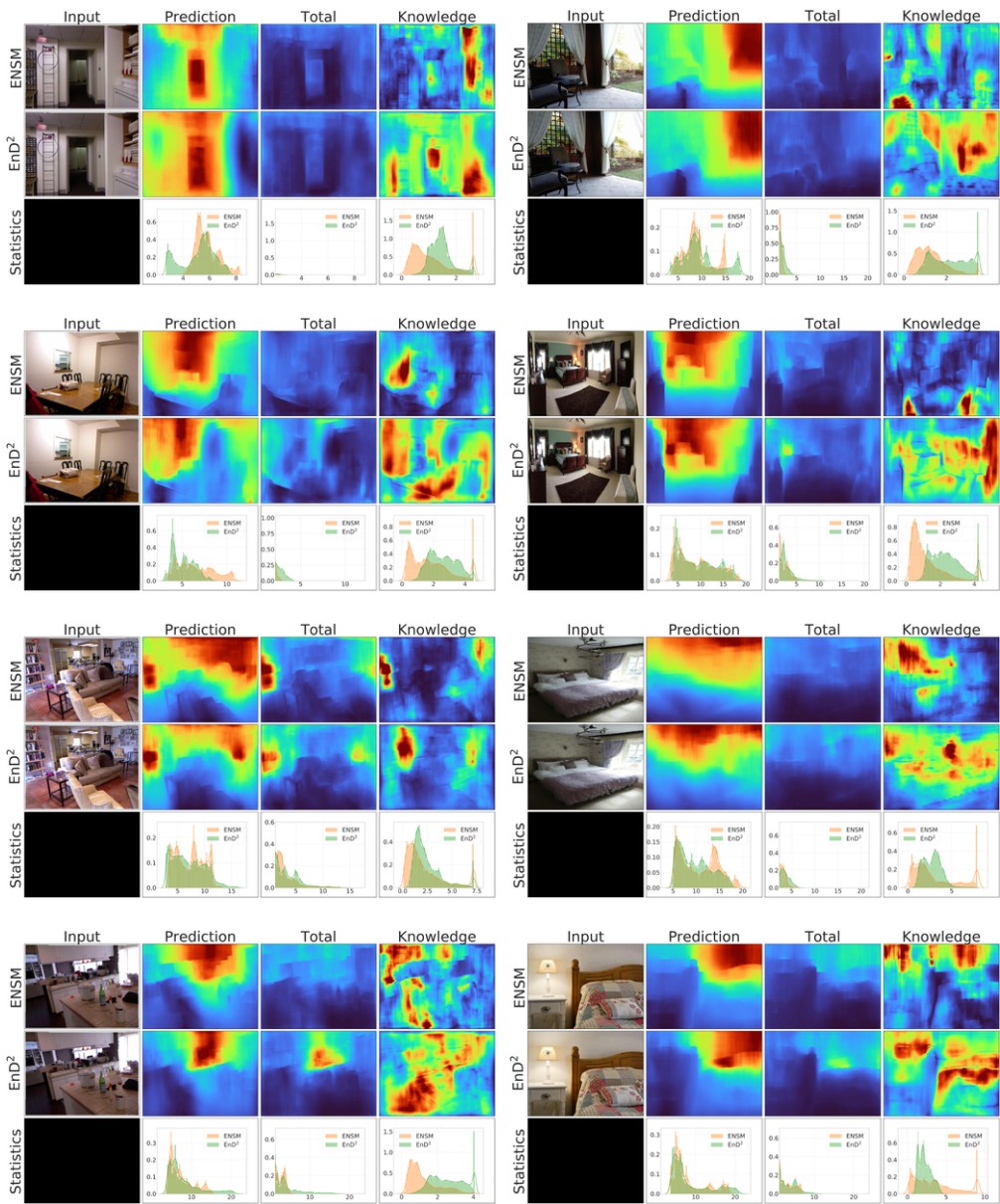

Figure 11: Uncurated comparison of ENSM vs EnD$^2$ models trained on KITTI behaviour on Nyu and LSUN-bed datasets (best viewed in color).

