# OpenReview forum: "Regression Prior Networks"
_ICLR.cc/2021/Conference — Reject_

### Official Review · AnonReviewer2 · 2020-10-21
**An nice paper based on incremental work**

**Rating:** 6
**Confidence:** 3

**Review:**

Summary of the Paper:

        This paper introduces regression prior networks. These are models that aim at capture predictive uncertainty, both epistemic and aleatoric, in the context of regression problems. Regression prior networks can also be used to compress an ensemble of predictors into a single model while keeping the benefits of the ensemble. That is, better predictive performance and uncertainty estimates. The method is validated on several problems from the UCI repository and compared with ensemble methods.

Specific details:

        I believe that this is a nice paper that illustrates an appealing method for uncertainty estimation in the context of neural networks. My main concern, however, is that it builds heavily on previous work. In particular, prior networks have already been proposed for classification and they have also been used to distill (compress) an ensemble. There is hence not much novelty here, only the extension to regression problems since, previously, only classification problems have been addressed. The use of prior networks for ensemble distillation is also not new. All this questions the novelty of the proposed approach.

        The extension to regression seems to follow very closely the work already carried out for classification. The only difference is that a Normal Wishart distribution is used instead of a Dirichlet distribution.

        The experiments carried out are extensive and consider different tasks involving prediction accuracy and out of distribution data detection. My main concern, however, is that no comparison is carried out with alternative methods to estimate prediction uncertainty such as those of Bayesian neural networks using variational inference or dropout. The authors should comment on the advantages of their method with respect to these techniques.

        The method proposed is also complicated and has several training parameters. The authors give specific values for them, but it is not clear the motivation for them or the sensitivity to their values.

        The paper is clearly written but heavily relies on previous work, making the reading difficult for someone who is not familiar with it. The paper is not self-contained.

        Summing up I believe that this could be an interesting contribution for the conference, suffering from a reduced amount of novelty.

---

> ### Author Response · Authors · 2020-11-13
> **Reply to Reviewer 2 Comments**
>
> Thank you for your comments! We will now address your comments point-by-point:
>
> 1. Regarding alternative Bayesian baselines:
>
> The approach we use to generate ensembles  - Deep Ensembles [4], are already the current go-to SOTA Bayesian approach to uncertainty estimation [1,2,3]. Approaches like Dropout, while also capable of generating ensemble, are shown to be consistently inferior. Variational Inference is typically even worse and has never been successfully scaled to complex tasks such as Depth Estimation, to our knowledge.
>
> Our favoured proposed approach - Ensemble Distribution Distillation for regression, allows us to take a SOTA DeepEnsemble (which is the baseline relative to which we compare) and distill it into a single model, generally preserving most of the ensemble’s gains. This allows us to replicate both the ensemble’s predictive performance as well as uncertainty measures at the computational and memory cost of a single model. Thus, suffer a minor reduction in predictive quality (and no loss in the quality of uncertainty estimates) for an M-fold (where M is the ensemble size) reduction in computational and memory cost relative to the ensemble baseline.
>
>        [1] Can you trust your model’s uncertainty? Evaluating predictive uncertainty under dataset shift.
>
>        [2] Pitfalls of in-domain uncertainty estimation and ensembling in deep learning.
>
>        [3] Deep ensembles: A loss landscape perspective.
>
>        [4] Simple and Scalable Predictive UncertaintyEstimation using Deep Ensembles
>
>
> 2. Regarding additional training parameters - Could you please be more specific, so that we could address your concerns in detail?
>
> 3. Regarding the difficulty of understanding the paper - Are there particular changes you would like us to implement which you think would make this paper more accessible?

---

> > ### Comment · AnonReviewer2 · 2020-11-16
> > **Response to Authors**
> >
> > Thank you for your reply.
> >
> > What I was missing is a high-level description of prior networks. A background section on already existing prior networks for e.g. classification could be nice to include.
> >
> > Regarding the parameters, I was referring to the gamma parameter in Eq. (8) and the beta parameter in Eq. (11).

---

> > > ### Author Response · Authors · 2020-11-20
> > > **Response to reviewer 2**
> > >
> > > Thanks for you for your comments!
> > >
> > > We explored the effect of gamma in the depth estimation setting in the appendix, table 11. The effect of beta is primarily to make the the expected (under Normal-Wishart) negative-log-likelihood be a tight bound to the NLL of the expected (vs expected NLL). High beta makes the bound tight and the training to be more accurate. We will add this this to the discussion and make it more clear.
> > >
> > > Regarding a discussion of Dirichlet Prior Networks - unfortunately, the space is rather limited to be able to discuss everything in detail. We will try to improve the discussion of RPNs such that it is more self-contained.

---

### Official Review · AnonReviewer3 · 2020-10-26
**Clarifications needed**

**Rating:** 6
**Confidence:** 4

**Review:**

This paper addresses interpretable uncertainty quantification for data driven models. In particular, the authors focus on a sub-class of methods known as Prior Networks and attempt to extend these methods to regression tasks as existing approaches address classification only. The author contribution is thus clearly stated and positioned w.r.t. prior arts and tackle a non-trivial issue.

In the classification setting, the Dirichlet distribution is pretty much the universal model for the parameters of multinomial distributions. For regression, i.e. continuous r.v., there is no such universal solution and the authors chose to focus on outputs that have a normal distribution. The parameters of this latter are assumed to be normal-Wishart. Although, the proposed method is de facto non-applicable to other types of distributions, it can be argued that this already covers a majority of situations.

The paper is rather well organized and seems technically sound. This said, a few mathematical details are missing and, most importantly, the experiments are not very convincing. These concerns, also with other minor remarks are detailed below, section by section.

sec 2.2

Maybe give the explicit definition of Z to clarify that is does not depend on network parameters.
The presence of the OOD loss term in (8) is a bit artificial as it boils down to regularizing because of the choice of beta. Is this choice systematic ?
In (9), how is p(y | mu, Lambda) computed ? Is it a T distribution ?

2.3

(12) lacks clarity : dataset is equal to an empirical distribution... Do you mean p hat is a sum of Dirac ?
What does phi represent ?

3
The acronym ENSM is not explained. I believe this corresponds to the deep ensemble.
The Prior Networks achieve a form of disambiguation but the quality of it is a bit disappointing compared to ENSM. In particular, data uncertainty raises quickly for out-of-domain inputs.

4

The presentation of the experimental protocol in 4 lacks clarity thereby impairing the interpretation of the results. The definition of the unconventional performance criteria [(Malinin et al. 2020] must be recalled (at least in an appendix).
In addition, as honestly mentioned by the authors, these datasets may not offer sufficiently rich problems to provide interesting comparisons. Besides, the way that OOD data is generated does not seem to necessarily produce inputs that are not covered by the in-domain distribution. Perhaps, the authors could use a "bad GAN" to obtain such data points, I mean a GAN where the generator and the discriminator would co-operate instead of being adversaries. If it converges, the generator would produce synthetic inputs that are easy to discriminate, thus far from true inputs.

5
While the dataset used in this section is more challenging, the experiment description is confusing. Again, performance criteria are not sufficiently explained and the general message becomes cryptic. Table 3 is overly complicated, I think RMSE is fairly enough to depict regression performances. Moreover, the definition of some columns are missing.
In Table 4, the performances of the methods seem quite unstable. For example, NWPN works fairly well for a given dataset configuration for one knowledge uncertainty criterion but fails miserably using another criterion on the same data.
On Fig 3, from what dataset are these image coming from ? Why are these or that object presumably "unknown" to the model ?
I think the whole section deserves some re-writing.

Final remark : there are a few English mistakes that should be wiped out.

---

> ### Author Response · Authors · 2020-11-13
> **Reply to Comments of Reviewer 3**
>
> Thank you for your detailed comments! Allow us to address them:
>
> SEC 2.2 -
>   A. Z is indeed a constant that doesn't depend on the parameters of the model. We will make this clear.
>   B. I'm afraid that this isn't the case. The OOD loss doesn't regularise the choice of beta. Rather the OOD loss is supposed to inform the model of regions beyond which it has no understanding of the data. Clearly, this requires one to decide on and choose an OOD dataset, which is non-trivial.
> C. p(y | mu, Lambda) represents a Normal distribution sampled from the Normal-Wishart.
>
> SEC 2.3 - Yes, this is what I mean - the dataset can be seen an empirical distribution to which we minimise KL, or equivalently, maximise likelihood. Phi represents the parameters of the model into which we are distribution-distilling the ensemble.
>
> SEC 3.  ENSM is the Deep Ensemble. Respectfully, the behaviour of the estimates of data uncertainty out of domain is not relevant - data uncertainty is only important in-domain. Indeed, we cannot give any guarantees on the behaviour of data uncertainty in the OOD region. What we actually care about is that estimates of *knowledge uncertainty* increase as we move further out of domain, which is the case (though perhaps not so easy to see from the picture). We will update the picture to make this clearer.
>
> SEC 4
> A. We will make the experimental protocol clearer in this section. We will add the description of the Prediction Rejection Ratio in the appendix. It shows what part of the best possible error-detection performance our algorithm covers.
>
> B. UCI datasets are very common datasets for evaluation in related works, that’s why we decided to add them despite their simplicity.
>
> C. To obtain train-OOD-data for RKL, we used factor analysis with increased noise and latent variance. This is a simple generative model. We trained it on in-domain data and added noise to the latent variables to generate out-of-domain examples for RKL. This generative model is simple and appropriate for table data, while GANs are not usual for table data. Also, UCI datasets have few examples and small feature spaces, therefore it could be hard to train GANs on them.
>
> D. For the evaluation of OOD-detection performance, we took parts of other UCI datasets as OOD data. We made sure that the OOD-data comes from different domains and feature distributions are different. We felt that this was the best we could, as, to the best of our knowledge, there has been no established research on OOD detection for tabular datasets.
>
> SEC 5
>
> Performance metrics in table 3 are usual for Monocular Depth Estimation. They describe model performance from different sides and are usually shown in all papers on this topic. A good description of these metrics can be found in the original Monocular Depth Estimation paper “Depth Map Prediction from a Single Image using a Multi-Scale Deep Network” by Eigen et al., in section 4.3.
>
> Delta 1,2,3 shows a percent of predictions such that the maximum of two fractions: (a) between predictions and targets, (b) between targets and predictions is less than corresponding thresholds: 1.25, 1.25^2, and 1.25^3. Rel stands for absolute relative error and log10 for RMSE between logarithms of predictions and targets. These losses show different properties of the model: deltas help to understand confidence intervals of the model, Rel shows the ratio between prediction error and target, and log10 shows error in the log-space.
>
> We will add the definition of these metrics to the text and  attempt to simplify table 3 as much as possible.
>
> We fully understand where you are coming from regarding table 4 and figure 3. We will rewrite this section and make it more understandable, it was hard to fit everything into a given space.
>
> Regarding Table 4 and the behaviour of the NWPN (RPN+RKL) model - we hypothesise this is the result of the interaction between the in-domain and OOD training data. It was very hard to get the models to appropriately train. Likely because discrimination between ID/OOD is a very global task (global scene understanding), while depth estimation requires more local data. The tasks are therefore anti-correlated in training. In contrast, EnD$^2$ doesn't suffer from the same problems and only relies on ID training data.
>
> Thus, what we aim to show is that: 1) EnD$^2$ can appropriately replicate and surpass the ensemble's OOD performance. 2) NWPN (RPN+RKL) can sometimes do near-perfect OOD detection, but isn't as reliable in this particular task with this choice of OOD data.
>
>
> In Figure 3 the left image is from KITTI and the right image is from NYU datasets. Using these images we aim to show that error of a prediction correlates with increased uncertainty of the model. Additionally, we wanted to show how the uncertainties of the ensemble and EnD$^2$ model compare, and we can see that the EnD$^2$ model consistently yields higher uncertainties, as it over-estimates the support of the ensemble.

---

### Official Review · AnonReviewer4 · 2020-10-28
**A simple extension of Prior networks models to regression.**

**Rating:** 5
**Confidence:** 4

**Review:**

This paper extends Prior networks models, previously introduced for classification, to regression problems.  Prior networks are neural networks whose main target is to "modelling uncertainty in classification tasks by emulating an ensemble using a single model".  Standard Prior networks models output the parameters of a Dirichlet probability distribution. This Dirichlet probability distribution then defines a distribution over categorical probability distributions over the different classes. This hierarchical approach allows to better capture uncertainty. The presented approach extends this framework to regression tasks. So, instead of returning the parameters of a Dirichlet distribution, it returns the parameters of a Normal-Wishart distribution, which then defines a probability distribution over Normal distributions, and, in turn, each Normal distribution defines a probability distribution over the value of the target variable.


Pros:
* The presented approach is sound and addresses a relevant problem, which is modelling uncertainty for regression problems.
* A method for distilling an ensemble model into a single model while maintaining accuracy is also proposed.
* The proposed approach does not incur in computational and memory overheads like standard deep ensembles.
* This work properly approaches technical difficulties (such as employing numerical stable precision parametrizations of the Normal-Wishart distribution) that arise in this kind of problems.

Cons:
* The presented approach does not introduce any novel idea or insight. It's a relatively simple extension of a previously published method.
* The empirical results do not show a clear advantage of the presented approach wrt previously published proposals.
* The advantage of having a small computational and memory overhead is not properly evaluated with other proposals which also have a small  computational and memory overhead [1] (although this proposal has not been defined for regression problems, the adaptation to regression is as simple as the adaptation of the DeepEnsembles models employed in this work).


I can not recommend the acceptation of this paper because I find the originality of the work quite limited. Although the extension of prior networks to regression task is mot really straightforward because of technical issues related to the problem of learning the parameters of a Normal-Wishart distribution. The general strategy to do that exactly matches the previous steps employed when introducing prior networks.  In consequence, this work does not provide any new relevant insight into the problem of modelling uncertainty and learning models with well-calibrated predictions.


Minor comments:
- Eq (14): T parameter is not defined. Temperature?
- Typo at the end of Page 5: [-25,20] --> [-25,-20]
- ENSM is defined after Table 1.
- Fix the following reference:
Andrey Malinin and Mark JF Gales. Reverse kl-divergence training of prior networks: Improved uncertainty and adversarial robustness. 2019.

Post-rebuttal:  I thank  the authors' effort for the improvement of the manuscript following the comments of the different reviewers. I think the overall quality of the paper has really improved. But, after many thoughts, I still think there is a limited novelty in this paper. I have increased my score to 5. But I can not recommend this paper for publication.

  adding baseline models to the paper and missing citations. I do think this improves the overall paper by a lot. As mentioned already in my paper, I do believe this is a nice idea and executed well, even though novelty might be limited. I am keeping my score and recommending an accept.

[1]  Wen, Y., Tran, D., & Ba, J. (2020). BatchEnsemble: an Alternative Approach to Efficient Ensemble and Lifelong Learning. arXiv preprint arXiv:2002.06715.

---

> ### Author Response · Authors · 2020-11-13
> **Reply to Reviewer 4 Comments**
>
> Thank you for your review! Please allow us to address your concerns:
>
> 1. Regarding empirical results:
>
>  Could you please elaborate what you would see as a clear advantage? In terms of inference-time compute and memory the M-fold (where M is the ensemble size) advantage over Ensembles is clear. In terms of predictive performance - we outperform single models, and get close to the ensemble. Replicating the ensemble’s predictive performance completely is an upper bound. In terms of OOD Ensemble-Distribution Distilled  RPNs outperform the ensemble. If there some specific comparison you would like us to provide which would convince you?
>
> 2. Regarding BatchEnsemble:
>
> BatchEnsembles are interesting, however an efficient implementation of BatchEnsembles is non-trivial and there is no available code in pytorch (The original work was done in Edward). A naive implementation would be as expensive during inference as DeepEnsembles, if not more so, as it may require a larger ensemble to reach the same performance. If you insist, we will explore this approach, but this will likely be infeasible within the time-frame of the rebuttal period.
>
> P.S. Thank you for finding the minor errors. We will fix them.

---

> > ### Comment · AnonReviewer4 · 2020-11-17
> > **Reply to Authors comments**
> >
> > Thank you for reply!
> >
> > I agree deep ensembles can be considered as an upper bound. My point is that there are alternative methods like BatchEnsembles, Rank-1 BNNs, SNGP, etc. (this repo https://github.com/google/uncertainty-baselines contains the references and high-quality open source implementations of all of them) which could be easily adapted for regression and which have much lower time and memory complexity than ensemble methods. In my opinion,  at least one of them should be considered here as relevant baseline, because  only comparing wrt deep ensembles gives the impression that your method is the only available alternative that provides  a big reduction in time and memory complexity  wrt deep ensembles. I think it is fair to show (or at least discuss) that there are other approaches that can be employed here to strongly reduce the memory and time complexity of deep ensembles.
> >
> > The lack of novelty of this paper, as acknowledged by other reviewers, puts much more pressure in the empirical evaluation. As I said before, there are well-established prior works which directly address the high memory and time complexity of deep ensembles that can be easily adapted to regression and which, in my opinion, should be considered by this work.

---

> > > ### Author Response · Authors · 2020-11-20
> > > **Response to R4**
> > >
> > > We are very happy to provide a discussion of various alternative approaches to making ensembles computationally cheaper. Indeed, we’ve found 2 papers on a similar approach to distilling an ensemble into a single model [1,2] by having multiple output heads, where each head is meant to replicate the behaviour of a particular ensemble member. We believe that this is as close a baseline as we can make - it is almost identical in compute to EnD^2, it's also a distillation approach, and it also attempts to preserve ensemble diversity. We have provided these results  in TABLE 3 and TABLE 4 of the updated manuscript. Generally, this works a little better for predictive quality than EnD^2, and worse for OOD detection. Results on Kitti for MDD are not ready yet, but are being calculated.  Do you find these results sufficient?
> > >
> > > Regarding BatchEnsembles - we’ve had a closer look, and we currently don’t actually understand how it is cheaper *at run time*. While it is true that a BatchEnsemble model has about as many parameters as a single model *on disk*, at *run time* it trades of increased use of GPU memory (batch is replicated) for efficient use of said GPU. Thus, it may be faster than sequential evaluation of an explicit ensemble, but it certainty is not more memory efficient at run time. We will certainly cite, mention and discuss this range of works. However, implementing it is non-trivial -the libraries you’ve sent us are in Tensorflow, not Pytorch, so we cannot directly carry them over.
> > >
> > > If you insist, we CAN promise to implement BatchEnsembles and add this into the camera ready paper (if this paper is accepted). We would definitely keep this promise, as it would be quite embarrassing to make it publicly and then break it.
> > >
> > > [1] HYDRA: PRESERVING ENSEMBLE DIVERSITY FOR MODEL DISTILLATION (Tran et al).
> > > [2] Ensemble Approaches for Uncertainty in Spoken Language Assessment, Wu et al, 2020, Interspeech.

---

> > > > ### Comment · AnonReviewer4 · 2020-11-23
> > > > **Reply to Response to R4**
> > > >
> > > > Thanks again for your nice replay!
> > > >
> > > > I think my concerns, in terms of comparison, with other related works are already addressed. The comparison with multi output heads is also reasonable and enough for me. Even though, I think it would be worth to also include the discussion of other methods like BatchEnsembles.
> > > >
> > > > Regarding novelty and relevance of the work, I still have the concerns I rose in my original review. But, I promise I will give new thoughts in the light of other reviewers and all your responses.
> > > >
> > > > Thanks for the fruitful discussion.

---

> > > > > ### Author Response · Authors · 2020-11-23
> > > > > **Thank you for the stimulating discussion!**
> > > > >
> > > > > We've updated the paper and are about to make a post describing all the updates. We've added a discussion about BatchEnsembles and the Multi-head distillation method.
> > > > >
> > > > > We appreciate the effort you put into this discussion!
> > > > >
> > > > > Many thanks,
> > > > > Authors

---

### Official Review · AnonReviewer1 · 2020-10-28
**Official Blind Review #1**

**Rating:** 6
**Confidence:** 4

**Review:**

Prior Networks (Malinin & Gales, 2018) use Dirichlet prior over categorical predictive distributions to distill ensembles for classification tasks. This paper extends Prior Networks to the regression setting by using a Normal-Wishart prior in order to attempt to match the predictive diversity. The authors define the model and loss terms including analytical derivation and evaluate their proposed approach with synthetic data, UCI datasets and monocular depth estimation.

_Strengths_:
- The paper is well-written and clearly structured.
- Most design choices are justified.
- Simple idea (in a good way!) which seemed to work well, shown by the evaluation.

_Weaknesses_:
- Most of the work seems to be heavily based on Prior Networks (Malinin & Gales, 2018). Even Section 2.1 seems to be exactly like the Subsection in the paper about Prior Networks. This paper mainly focuses on an extension to the regression task. Therefore, the contribution / novelty of this paper is incremental. However, I still think the authors did a good job to present a general distillation method for regression task. Therefore, I would consider the novelty a minor weakness.
- I am on the fence about specifying the OOD dataset for learning with the loss in Eq. 8. I believe it is difficult to decide what kind of model to use for generating the OOD dataset, thus, the model choice can lead to large differences in performance. This is not really discussed. Further, the models trained have more data available for training, I believe it is not quite fair to compare against models which only have been trained on in-domain-data.
- There are no comparisons to other approaches for distillation of regression tasks. I understand, that this paper wants to show a viable general approach for regression distillation, however, this work is not the first one to do so and therefore should consider existing work.

_Overall assessment_: For me, this paper is borderline. The weaknesses, especially the OOD dataset used for training and the lack of comparisons in the evaluation are concerns. However, I like the idea and the execution so therefore, I would recommend a weak accept (6).

_Detailed comments and questions_:
- OOD data: I have seen that you have an ablation for the degree of regularization on the OOD dataset. However, what about different OOD data? Why choose KITTY and not a different dataset? Were there any large difference in performance?
- Table 3: I notice that NLL performance of distilled models are better than the actual ensemble, how can this be?
- OOD detection for monocular depth estimation: Did you also trained the comparing models with the OOD data, e.g. DD?
- Comparing models: Have you consider comparing your model to other ones, e.g. [1, 2]? This could improve your paper and approach to show that it also consider existing work on regression distillation.

_Post-rebuttal_:
I really appreciate the authors adding baseline models to the paper and missing citations. I do think this improves the overall paper by a lot. As mentioned already in my paper, I do believe this is a nice idea and executed well, even though novelty might be limited. I am keeping my score and recommending an accept.

[1] Chen, G., Choi, W., Yu, X., Han, T. and Chandraker, M., 2017. Learning efficient object detection models with knowledge distillation. In Advances in Neural Information Processing Systems (pp. 742-751).
[2] Saputra, M.R.U., de Gusmao, P.P., Almalioglu, Y., Markham, A. and Trigoni, N., 2019. Distilling knowledge from a deep pose regressor network. In Proceedings of the IEEE International Conference on Computer Vision (pp. 263-272).

---

> ### Author Response · Authors · 2020-11-13
> **Reply to Review 1 comments.**
>
> Thank you for your review! Allow us to address your concerns on a point-by-point basis.
>
> REGARDING WEAKNESSES
>
> 1. We agree with your concerns regarding the choice of OOD dataset. Defining an appropriate one for classification tasks is already non-trivial - doing so is even more challenging for regression. This is why we place greater emphasis on Ensemble Distribution Distillation - it does not require an OOD dataset and yields superior predictive performance relative to RKL-trained Regression Prior Networks.
>
> We will use the extra page to present a discussion regarding difficulties of using an OOD dataset, and will shortly upload an updated manuscript.
>
> 2. With regards to regression distillation, we would like to point out that previous work has examined the distillation of a *single model  into a single model*. In our work we consider distillation of *an ensemble of probabilistic models into a single probabilistic model*. Limited prior work has examined this scenario, and it is difficult to provide a sensible baseline . We have attempted to do so through Ensemble Distillation (EnD), though it seems this approach also has its limitations.
>
> It is, in general, not entirely clear whether combining an ensemble of probabilistic models is better done as an arithmetic or geometric mixture. A full analysis of ensembles of probabilistic regression models deserves an investigation of its own. Furthermore, to our knowledge, probabilistic ensemble distillation for regression has been a generally under-explored area. If you could point us to a more appropriate baseline, we would be happy to consider it!
>
> We will add a detailed discussion of this issue into section 2.3 and upload an updated manuscript shortly.
>
> REGARDING DETAILED COMMENTS
>
> 1 We were limited in the compute we had available for this project and decided to focus on the ablation study we did, rather than swapping out OOD datasets. In general, for Depth Estimation, we would like to place greater emphasis on RPNs trained through EnD$^2$, rather than RPNs trained via RKL on OOD datasets.
>
> Indeed, one of the conceptual reasons for not further exploring choice of OOD datasets for RPN+RKL is that we believe (and show) that RPNs+EnD$^2$ to be the superior approach.
>
> We will clarify this point in an updated manuscript we will shortly upload.
>
> 2. We believe this is a result of the fact that the EnD$^2$ will overestimate the support of the ensemble (as a natural consequence of ML training). As a result, it will be less over-confident.
>
> 3. We didn’t. To be clear - we intended our main comparison for Depth Estimation to be Ensembles vs  EnD$^2$ . Note that RPNs trained via RKL on OOD data in section 5 suffer degraded predictive performance. On the other hand, RPNs trained via  EnD$^2$ show better predictive performance (relative to EnD, Single models and RPN+RKL).
>
> 4. : Thank you for pointing out this work. However, as previously stated, these papers consider the distillation of single model into single model, and thus cannot be used as a meaningful baselines. However we will cite them when discussing the nature of regression distillation and highlighting how our work is different.

---

### Author Response · Authors · 2020-11-13
**Addressing concerns regarding novelty**

Dear Reviewers,

All of you have expressed concerns regarding the novelty and originality of our work. We would like to address this issue and explain why we think this merits a paper.

In our interactions with other researchers, and especially with industrial ML practitioners, we noticed that many people thought that the correct extension of Prior Networks to regression tasks would be to take a non-probabilistic regression model and place a Normal distribution over the target variable. As is clear from our work, this is not correct. Thus, one of the main motivations for this paper was to address this common misunderstanding and show that the correct way to extend Prior Networks and Ensemble Distribution Distillation to regression tasks.

In order to convey our message as clearly as possible we explicitly structured the paper around the parallel between Dirichlet and Normal-Wishart Prior Networks to make it absolutely self-evident what the correct approach is. In this regard we seem to have succeeded a little too well, as all of you note how the extension is straightforward and incremental. We would respectfully ask you to consider that this extension is not as evident to the majority of the ML community as we make it seem in this work. Notably, since the publication of the original paper on Dirichlet Prior Networks (Malinin and Gales, 2018), to our knowledge, Prior Networks have not been extended to regression, despite the popularity of the approach for classification. Thus, the value of our work is in extending a powerful uncertainty estimation approach for classification to regression, resolving a common misconception, and clearly presenting the mathematical basis for this extension.

We address your remaining concerns on a point-by-point basis and will shortly upload an updated manuscript.

Sincerely,
Authors

---

### Public Comment · ~Alexander_Amini1 · 2020-11-16
**Some missing related work**

Thanks for submitting this work! In line with many of the reviewer comments regarding novelty, I was also wondering about the relation of the proposed contribution to published evidential deep learning (EDL) approaches [1,2]. Namely, published at NeurIPS this year, Deep Evidential Regression [1] also proposes learning a 1D Normal-Wishart distribution directly to infer representations of uncertainty specifically in the continuous regression domain as well (not classification). The proposed contribution presented here, like [1], also provides experimental results on nearly identical tasks from UCI and on monocular depth estimation. Also, note that deep evidential networks are structurally identical to prior networks (PN), with the only differences being in their respective objective functions (PN additionally require OOD data to train with, EDL does not). Given that a preprint of [1] appeared over a year ago and is now peer-reviewed/published, as well as the foundational work done in the classification domain [2] is over two years old now, I think it would be very helpful for the authors to cite these papers and and discuss their contributions relative to these works.

I also hope this will help orient reviewers to the context for this submission and perhaps to some contributions that may have been missed.

[1] Amini, Alexander, et al. "Deep evidential regression." Advances in Neural Information Processing Systems. 2020.

[2] Sensoy, Murat, et al. "Evidential deep learning to quantify classification uncertainty." Advances in Neural Information Processing Systems. 2018.

---

> ### Author Response · Authors · 2020-11-20
> **Response to A. Amini**
>
> Thanks for your comment!
>
> Regarding Evidential Deep Learning, particularly [2] - we think it is a rather elegant alternative interpretation for uncertainty estimation, rooted in Dempster-Schafer Theory of evidence, which yields a model which is structurally identical to a Dirichlet Prior Network. However, we are sceptical of the principal claim that this is a reliable single-model uncertainty estimation approach which doesn’t require OOD data or indeed any other approach to enforcing a particular behaviour for OOD inputs which has an understandable mechanism of action. However, that is only our opinion - a rigorous large-scale validation is necessary and clearly would be a useful future direction of investigation.
>
> Regarding your paper (congrats on getting into NeurIPS!). We see our work as being very much an extension and verification of the ideas proposed in [3]. We became aware of your work around the same time as we began ours, however, to the best of our knowledge (until your comment) it was a submission at ICLR2020. Furthermore, upon examination at the time, we determined that the loss function (expectation of square error given samples from the Normal-inverse-Gamma) had an error in the derivation (equations 7-9, derivation 7.1.2 eq. 22-23 in the appendix). As a result, we had no grounds on which to believe in the validity of the results. Looking at the ArXiv submission now - it has not been updated within the last year and still contains the error .
>
> With respect to the experimental setup - while we both use UCI (which is standard) and NYU Depth v2, our evaluations are quite different. We have used a standard architecture, provided detailed performance comparisons to baselines in depth estimation [4,6], and analysed the properties of several uncertainty measures via ROC-AUC against a range of OOD datasets.
>
> However, we have now found your new NeurIPS2020 version in the pre-proceedings (which were released after the ICLR2021 submission deadline), and we see that the mathematical error has now been fixed. In fact, an altogether different loss function (NLL of the student distribution) is used in addition to the evidence regularizer. The results are largely the same.
>
> We’ve implemented DER both as it is on ArXiv and as it is in the NeurIPS2020 pre-proceedings, and present the results in the next post. Unfortunately, a direct number-for-number comparison to your work is not possible, as there are no summary performance results for your model, and figure 4B contains a range for RMSE which is about 20 times smaller than what is reported in the depth estimation literature [4,6] (you’ve probably scaled something differently). Furthermore, you use a different OOD dataset which seems to be very easy to separate out, as all models achieve a ROC-AUC of about 0.99.
>
> We will add these results to our paper (omitting the old DER), and cite your work (and the original Evidential work), in our paper. We shall upload an updated version shortly.
>
>
> [1] Amini, Alexander, et al. "Deep evidential regression." Advances in Neural Information Processing Systems. 2020.
> [2] Sensoy, Murat, et al. "Evidential deep learning to quantify classification uncertainty." Advances in Neural Information Processing Systems. 2018.
> [3] Uncertainty Estimation in Deep Learning with Application to Spoken Language Assessment, Malinin, 2019. PhD Thesis
> [4] High Quality Monocular Depth Estimation via Transfer Learning (Alhashim & Wonka, 2018)
> [5] HYDRA: PRESERVING ENSEMBLE DIVERSITY FOR MODEL DISTILLATION (Tran et al).
> [6] https://paperswithcode.com/sota/monocular-depth-estimation-on-nyu-depth-v2 ,
> [7] Ensemble Approaches for Uncertainty in Spoken Language Assessment, Wu et al, 2020, Interspeech.

---

> > ### Author Response · Authors · 2020-11-20
> > **Experimental results**
> >
> > We initially tried to add table here, but unfortunately it turns out that the system ignores formatting and just dumps numbers. Results are presented in TABLE 3 and TABLE 4 of updated manuscript.
> >
> > For all versions of DER (2019 ArXiv and 2020 Neurips) we used a weight of 0.1 on the evidence regulariser. For DER 2020 we checked the implementation of the student NLL against the pytorch version and made sure that everything is correctly parameterised.
> >
> > The results show a few things.
> >
> > Firstly, the old version of DER (ArXiv 2019) doesn't work well, both in terms of predictive performance, and in terms of uncertainty estimation. Which expected, due to the error on the loss.
> >
> > Secondly, the new version of DER (NeurIPS 2020 preproceedings) works much better. In terms of predictive performance it is comparable to a single probabilistic DenseDepth model , though still with a minor degradation. In terms of OOD detection performance of models trained on NYU- it  yields rather competitive performance, marginally worse than the ensemble, and outperformed by EnD^2.
> >
> > Thirdly, on KITTI OOD detection the LSUN OOD data is OOD not only because it is indoors, but also because it represents images which are very close to the camera, relative to images seen in Kitti, which features a range of depths. Here, all models, except RPN + RKL, interpret the OOD data as being in-domain using measures of total uncertainty. Using measures of ensemble diversity (knowledge uncertainty), ensembles, RPN-RKL and EnD^2 are able to detect OOD images successfully. Notably, DER does not seem to be able to. The reason for this is that the evidence regulariser biases the DER model tol yield high evidence in regions of low absolute error, and low evidence in regions of high absolute error. As a result, regions which are closer (bottom half of kitty images) always have higher evidence. As LSUN OOD is very close to the camera, the DER model yields high evidence.

---

### Author Response · Authors · 2020-11-20
**New results in manuscript**

Dear All,

We tried to provide new results in the open review system, but it turns out that it is very poor (and inconsistent across write/preview and what it actually posts) at representing tables. Thus, we have updated TABLE 3 and TABLE 4 in the manuscript, where we have added results for:

Deep Evidential Regression (ArXiv 2019 version) (will only be displayed during rebuttal)

Deep evidential Regression (NeurIPS 2020 pre-proceedings version) [2]

Mixture Density Distillation (only NYU Depth V2 so far, KITTI still training...) [3,4]

NOTE - we are still in the process of updating the text. This update is purely intended to demonstrate updated results.


[1] Amini, Alexander, et al. "Deep evidential regression." ArXiv. 2019 (version 1)

[2] Amini, Alexander, et al. "Deep evidential regression." Advances in Neural Information Processing Systems. 2020.

[3] HYDRA: PRESERVING ENSEMBLE DIVERSITY FOR MODEL DISTILLATION (Tran et al).

[4] Ensemble Approaches for Uncertainty in Spoken Language Assessment, Wu et al, 2020, Interspeech.

---

### Author Response · Authors · 2020-11-23
**Updated Manuscript Text**

Dear Reviewers,

As per the reviewers' and public comments, we have provided additional comparisons to Deep Evidential Regression and Mixture-Density Distillation, as we have described in our previous post. Now we have made changes to the text to reflect the reviewers' other comments. We hope that these changes sufficiently address all of your concerns.

We describe the changes section by section:

Introduction (minor changes)
1. Modified second paragraph to mention evidential approaches
2. Modified final paragraph to correctly point to roots of idea

Regression Prior Networks (lots of changes)
1. Sections styles changed for extra space
2. Added clarifications into discussions of RKL loss function, including role of OOD data, and effect of beta.
3. Added a discussion of EnD and MD-EnD to ensemble distribution distillation section
4. Added a final "Related work" section, which discusses Deep Evidential Regression and efficient ensemble methods (batch ensemble.

Synthetic Experiment
1. X-axis range in images widened, so that it is clear from figure C that knowledge uncertainty rises sharply

UCI Experiments (minor tweaks)
1. Section reworked, experimental protocol clarified.
2. Added reference to appendix which C3 which discusses PRR

Monocular Depth Estimation (SIGNIFICANTLY reworked for clarity)
1. Clarified the depth estimation performance metrics and made a forward reference to appendix D1, which they are described.
2. Added experiments on DER and MD-EnD to both tables 3 and 4
3. Restructured discussion of OOD experiments. Behaviours explains.
4. Added forward reference to an additional set of OOD detection experiments in the appendix.
5. Added forward reference to examples of IN/OOD data so that it is easy to see *exactly* what the models are trying to discriminate between.
6. Expanded discussion about the difficulty of choosing appropriate OOD data, which highlights that EnD^2 is the superior approach, as it doesn't suffer from this difficulty.

Conclusion (Minor clarification at the end)



We thank all the reviewers for their effort!

Sincerely,
Authors

---

### Author Response · Authors · 2020-11-24
**Final revision**

We added a final revision to fix a mistake in eq. 13 and fix a few typos.

---

### Decision · Program_Chairs · 2021-01-07
**Final Decision**

**Decision:**

Reject

**Comment:**

This paper presents a useful contribution to the growing literature on uncertainty estimation with deep learning. The review process has significantly helped with strengthening this paper, specifically with the concerns about novelty and sufficient comparisons to existing work. I hope you will continue to improve this work for submission to a future venue.